# TOWARDS COMPRESSIVE AND SCALABLE RECURRENT MEMORY

## ABSTRACT

Transformers face a quadratic bottleneck in attention when scaling to long contexts. Recent approaches introduce recurrent memory to extend context beyond the current window, yet these often face a fundamental trade-off between theoretical principles and practical scalability. To address this, we introduce ***Elastic Memory***, a novel memory architecture grounded in the HiPPO framework for online function approximation. Elastic Memory treats historical sequence as samples from continuous signals, applying optimal online compression to encode them into a fixed-size memory state. For retrieval, we propose a flexible *polynomial sampling* mechanism that reconstructs a history summary from this compressed state. Elastic Memory consistently outperformed baselines on long-context (32k+) datasets across three domains. With equal parameters, it beat Memorizing Transformer by 16x memory and outperformed Melodi at all memory sizes, even when Melodi had 30% more parameters. When scaling model size, Elastic Memory stayed ahead of all baselines and was significantly faster than Melodi at 4x size. Furthermore, its decoupled design allows for injecting inductive biases at test-time to boost performance. [1]

## 1 INTRODUCTION

The central challenge in extending the context length of language models lies in efficiently managing an ever-growing history while preserving information fidelity. Existing approaches to this problem often present a fundamental trade-off between theoretical principles and practical scalability. On one hand, paradigms like associative memory (Schlag et al., 2021; Munkhdalai et al., 2024) offer a theoretical foundation, framing memory updates as an online optimization of key-value associations. However, they face a critical scalability bottleneck: their memory capacity is rigidly coupled with model dimensions such as the number of heads and head dimension, precluding performance improvements by simply expanding the memory size. On the other hand, methods based on external memory (Wu et al., 2022) or summary tokens (Bulatov et al., 2022; Chen et al., 2025) provide practical scalability, as the memory size is a configurable hyperparameter. Yet, these approaches are often rooted in heuristic designs and lack a formal answer to what constitutes optimal memory compression, potentially leading to suboptimal results. This dilemma highlights the need for a new memory paradigm that unifies the rigor of a principled formulation with the flexibility of a scalable architecture. We argue that the HiPPO (High-order Polynomial Projection Operators) framework (Gu et al., 2020) provides the theoretical cornerstone for such a paradigm, reframing the problem of memory from heuristic compression to a principled problem of online function approximation.

In this work, we introduce Elastic Memory, a novel memory architecture that operationalizes the HiPPO theory to resolve the aforementioned trade-off. Elastic Memory treats key-value sequences as sampling points from continuous signals and applies HiPPO for a mathematically optimal, incremental compression into a fixed-size memory state. For retrieval, we propose a flexible *polynomial sampling* mechanism that reconstructs a history summary from this compressed state, which is then seamlessly integrated into the attention mechanism. Our comprehensive experiments validate the effectiveness of this design. Across three diverse long-document (32k+) datasets, Elastic Memory achieves state-of-the-art performance. It exhibits great memory efficiency. With the same model parameters, our model outperforms the Memorizing Transformer, a classic memory model, with

---

[1] Code will be released at `anonymous.url.com`.

16x larger memory. It also consistently surpasses the SOTA Melodi model, even when Melodi is allocated twice the memory. This lead holds even when Melodi has 30% more trainable parameters. This advantage persists as we scale the model size, where Elastic Memory continues to outperform all baselines while being 50% faster than Melodi at the 4x model scale.

## 2 METHOD

In this section, we present our model, Elastic Memory. We begin by providing the necessary theoretical background on the HiPPO framework (Section 2.1), which forms the mathematical foundation of our approach and summarizes the theoretical foundations established by Gu et al. (2020). To ensure clarity, Equations 1–4 in Section 2.1 are adapted from prior literature. We then detail our original contributions: the memory update mechanism (Section 2.2), based on parallelized block-level HiPPO compression (Equations 5–6), and the memory retrieval mechanism (Section 2.3), which leverages a novel polynomial sampling technique (Equations 7–8). The complete forward pass is summarized in Algorithm 1.

### 2.1 BACKGROUND: THE HiPPO FRAMEWORK

**Memory as Function Approximation.** The theoretical cornerstone of our work is the HiPPO (High-order Polynomial Projection Operators) framework (Gu et al., 2020), which addresses the fundamental problem of incrementally representing a cumulative history of a signal within a fixed-size state. This challenge has been a long-standing issue in sequence modeling, originally highlighted in the context of continuous-time signals and recurrent neural networks. HiPPO provides a principled solution by reframing memory from an engineering problem of caching or heuristic summarization to a formal problem of online function approximation. The core idea is to continuously compress an input history signal, conceptualized as a function $f(t)$, by optimally projecting it onto a basis of polynomials. This projection yields a set of coefficients that serve as a compact, fixed-size representation of the entire history. While this theory originates from a continuous-signal perspective, its principle of optimal, incremental compression provides a powerful foundation for tackling the information bottlenecks observed in modern language model memory mechanisms. This function approximation viewpoint offers a mathematically rigorous path for designing memory systems, and we now introduce the technical details of the HiPPO framework.

**Mathematical Formulation of HiPPO-LegS.** The HiPPO framework formally casts the problem of maintaining memory as finding a polynomial $g^{(t)}$ of degree less than $N$ that best approximates the history signal $f_{\leq t}$. This is framed as a continuous optimization problem defined by a time-varying measure $\mu^{(t)}$, which assigns importance to past events. Our work specifically leverages the HiPPO-LegS (Scaled Legendre) variant, which is defined by two key mathematical constructs.

First, HiPPO-LegS employs a scaled measure that uniformly weights the entire history from the beginning up to the current time $t$. This measure induces an inner product on the space of functions defined as:

$$\langle f, g \rangle_{\mu^{(t)}} = \int_0^t f(x)g(x)\frac{1}{t}dx. \tag{1}$$

The corresponding optimization objective is to minimize the squared $L_2$ norm of the approximation error: $\|f_{\leq t} - g^{(t)}\|^2_{L_2(\mu^{(t)})}$. This uniform weighting over the expanding interval $[0, t]$ is a crucial property that endows the resulting memory system with robustness to varying timescales.

Second, the optimal solution to this problem is found by projecting the signal onto an orthonormal basis. HiPPO-LegS constructs this basis from the classical Legendre polynomials $\{P_n(x)\}_{n=0}^{N-1}$. These polynomials are first affinely transformed to be orthogonal on the interval $[0, t]$, and then normalized to form an orthonormal basis $\{g_n^{(t)}(x)\}_{n=0}^{N-1}$, where $g_n^{(t)}(x) = \sqrt{2n+1}P_n(\frac{2x}{t} - 1)$.

The memory state is then defined as the $N$-dimensional coefficient vector $c(t) \in \mathbb{R}^N$ obtained by projecting the history signal onto this basis:

$$c_n(t) = \langle f_{\leq t}, g_n^{(t)} \rangle_{\mu^{(t)}} = \int_0^t f(x)g_n^{(t)}(x)\frac{1}{t}dx. \tag{2}$$

---

**Algorithm 1** Elastic Memory Attention Block Forward Pass

---

**Require:** Current input block $\mathbf{H}_{\text{curr}} \in \mathbb{R}^{L \times D_{\text{model}}}$; previous memory states $\mathbf{C}_{i-1}^{(K)}, \mathbf{C}_{i-1}^{(V)} \in \mathbb{R}^{N \times D}$; block index $i$.

**Ensure:** Output for current block $\mathbf{O} \in \mathbb{R}^{L \times D_{\text{model}}}$; updated memory states $\mathbf{C}_i^{(K)}, \mathbf{C}_i^{(V)} \in \mathbb{R}^{N \times D}$.

1: $\mathbf{Q}_{\text{raw}}, \mathbf{K}_{\text{raw}}, \mathbf{V}_{\text{curr}} \leftarrow \text{Linear}(\mathbf{H}_{\text{curr}})$           ▷ Project inputs to raw Q, K, V

2: $\mathbf{Q}_{\text{curr}}, \mathbf{K}_{\text{curr}} \leftarrow \text{ApplyRoPE}(\mathbf{Q}_{\text{raw}}, \mathbf{K}_{\text{raw}})$           ▷ Apply positional encodings
                                                      ▷ *1. Memory Retrieval*

3: Retrieve *reconstruction matrix* $\mathbf{R}_{i-1}$ from precomputed bank.

4: $\mathbf{K}_{\text{mem}}, \mathbf{V}_{\text{mem}} \leftarrow \mathbf{R}_{i-1}\mathbf{C}_{i-1}^{(K)}, \quad \mathbf{R}_{i-1}\mathbf{C}_{i-1}^{(V)}$           ▷ Eq. 8
                                              ▷ *2. Trapezoidal Attention*

5: $\mathbf{K}_{\text{aug}} \leftarrow [\mathbf{K}_{\text{mem}}, \mathbf{K}_{\text{curr}}]; \mathbf{V}_{\text{aug}} \leftarrow [\mathbf{V}_{\text{mem}}, \mathbf{V}_{\text{curr}}]$

6: $\mathbf{S} \leftarrow \frac{\mathbf{Q}_{\text{curr}}\mathbf{K}_{\text{aug}}^T}{\sqrt{D}}$

7: $\mathbf{O}_{\text{att}} \leftarrow \text{softmax}(\mathbf{S} + \mathbf{M})\mathbf{V}_{\text{aug}}$           ▷ Apply trapezoidal mask $\mathbf{M}$
                                                      ▷ *3. Memory Update*

8: Retrieve *state transition matrix* $\mathbf{P}_i$ and *HiPPO kernel* $\bar{\mathbf{K}}_i$ from precomputed bank.

9: $\mathbf{C}_i^{(K)}, \mathbf{C}_i^{(V)} \leftarrow \mathbf{P}_i\mathbf{C}_{i-1}^{(K)} + \bar{\mathbf{K}}_i\mathbf{K}_{\text{raw}}, \quad \mathbf{P}_i\mathbf{C}_{i-1}^{(V)} + \bar{\mathbf{K}}_i\mathbf{V}_{\text{curr}}$           ▷ Eq. 6

10: $\mathbf{O} \leftarrow \text{Linear}(\mathbf{O}_{\text{att}})$           ▷ Project output

11: **return** $\mathbf{O}, \mathbf{C}_i^{(K)}, \mathbf{C}_i^{(V)}$

---

This vector $c(t)$ completely characterizes the optimal polynomial approximation $g^{(t)}(x) = \sum_{n=0}^{N-1} c_n(t)g_n^{(t)}(x)$, thus serving as a compact, fixed-size representation of the entire history.

**From Continuous ODE to Discrete Recurrence.** A direct computation of the coefficients $c(t)$ via the integral in Eq. 2 at each timestep is computationally prohibitive, as it requires iterating over the entire history. The central insight of the HiPPO framework is that the evolution of this coefficient vector is governed by a linear Ordinary Differential Equation (ODE), enabling an efficient, incremental update (see Appendix E.1 for the complete derivation). For HiPPO-LegS, this ODE is given by:

$$\frac{d}{dt}c(t) = -\frac{1}{t}\mathbf{A}c(t) + \frac{1}{t}\mathbf{B}f(t), \tag{3}$$

where $\mathbf{A}, \mathbf{B} \in \mathbb{R}^{N \times N}$ are the HiPPO matrices, which are fixed and mathematically derived from the properties of Legendre polynomials, not learned. To apply this continuous-time model to discrete sequences such as text, the ODE must be discretized. Using a standard method such as the Zero-Order Hold (ZOH), which we use in our implementation for its stability, this process converts the continuous dynamics into a discrete recurrence relation that updates the memory state $c_k$ at step $k$ based on the state at step $k-1$ and the current input $f_k$:

$$c_k = \bar{\mathbf{A}}_k c_{k-1} + \bar{\mathbf{B}}_k f_k. \tag{4}$$

Here, $\bar{\mathbf{A}}_k$ and $\bar{\mathbf{B}}_k$ are the discretized state matrices, which depend on the step $k$ due to the time-varying nature of the ODE. This recurrence provides a mechanism to update the memory with constant computational and memory complexity at each step, making it a highly efficient building block for long-sequence models.

## 2.2 MEMORY UPDATE: ONLINE FUNCTION APPROXIMATION

**Compressing Over Time.** The discrete recurrence in Eq. 4 provides a highly efficient mechanism for updating a memory state. In our Elastic Memory framework, we integrate this mechanism directly into the Transformer architecture. Instead of treating the entire hidden state as a single signal, we apply the HiPPO compression to the Key ($\mathbf{K}$) and Value ($\mathbf{V}$) sequences. Specifically, we treat each feature dimension of the raw $\mathbf{K}$ and $\mathbf{V}$ projections as an independent one-dimensional signal evolving over the sequence length. It is crucial to note that the Key sequences compressed by HiPPO are those *prior* to the application of rotational position embeddings (RoPE), as the function approximation framework is designed to operate on the underlying signal.

**From Recurrent Update to Parallel Block Update.** A naive, token-by-token application of the recurrence in Eq. 4 within a block of length $L$ would be inefficient due to its sequential nature. To enable parallel processing, we transform the serial recurrence into an equivalent block-level update. By unrolling the recurrence over a block of inputs $\{f_0, f_1, \ldots, f_{L-1}\}$, the final state $c_{L-1}$ can be expressed as a linear combination of the initial state $c_{-1}$ and all inputs within the block (see Appendix E.3 for the detailed derivation):

$$c_{L-1} = \left( \prod_{k=L-1}^{0} \bar{\mathbf{A}}_k \right) c_{-1} + \sum_{j=0}^{L-1} \left( \prod_{k=L-1}^{j+1} \bar{\mathbf{A}}_k \right) \bar{\mathbf{B}}_j f_j. \tag{5}$$

This unrolled equation can be reframed as a single parallel operation. For the $i$-th block of the input sequence, with initial memory state $\mathbf{C}_{i-1}$ and input features $\mathbf{F}_i = [f_0, \ldots, f_{L-1}]^T$, the updated memory state $\mathbf{C}_i$ is given by:

$$\mathbf{C}_i = \mathbf{P}_i \mathbf{C}_{i-1} + \bar{\mathbf{K}}_i \mathbf{F}_i. \tag{6}$$

Here, $\mathbf{P}_i = \prod_{k=(i+1)L-1}^{iL} \bar{\mathbf{A}}_k$ is the state transition matrix that evolves the memory state across the entire block. The term $\bar{\mathbf{K}}_i \in \mathbb{R}^{N \times L}$ is a linear operator we term the *HiPPO kernel*, whose columns are composed of the products of $\bar{\mathbf{A}}_k$ and $\bar{\mathbf{B}}_k$ matrices from Eq. 5. Crucially, due to the time-varying nature of the HiPPO-LegS ODE, the discretized matrices $\bar{\mathbf{A}}_k$ and $\bar{\mathbf{B}}_k$ depend on the absolute position $k$. Consequently, the state transition matrix $\mathbf{P}_i$ and the HiPPO kernel $\bar{\mathbf{K}}_i$ are unique for each block position $i$.

**Efficient Parallel Update via Precomputation.** While Eq. 6 enables parallel computation within a block, the on-the-fly calculation of the position-dependent matrices $\mathbf{P}_i$ and $\bar{\mathbf{K}}_i$ would introduce significant computational overhead. To achieve maximum efficiency, we employ a precomputation strategy. Before training, we precompute and cache the state transition matrices and HiPPO kernels for all block positions up to a predefined maximum context length. During the forward pass, the memory update for the $i$-th block becomes a simple and highly efficient two-step process: (1) retrieve the precomputed matrices $\mathbf{P}_i$ and $\bar{\mathbf{K}}_i$; (2) execute the block update in Eq. 6 using two parallel matrix-multiplication operations. This strategy transforms the theoretically sequential HiPPO recurrence into a fully parallelizable block-level operation that can be efficiently executed on modern hardware, all while preserving the mathematical exactitude of the original HiPPO formulation.

## 2.3 MEMORY RETRIEVAL: POLYNOMIAL SAMPLING

**Precomputed Reconstruction Bank.** The retrieval of historical information is grounded in the reconstruction property of the HiPPO framework (see Appendix E.4 for the detailed derivation). The memory state $\mathbf{C}_{i-1}$, which holds the coefficients of the optimal polynomial approximation after processing $i-1$ blocks (total length $t = (i-1)L$), can be used to reconstruct the historical signal at any set of chosen sample points. This reconstruction is a linear operation. To maximize efficiency, we precompute a *Reconstruction Bank*, which is a collection of position-dependent reconstruction matrices $\{\mathbf{R}_0, \mathbf{R}_1, \ldots\}$. Each matrix $\mathbf{R}_i \in \mathbb{R}^{L_{\text{mem}} \times N}$ is designed to sample $L_{\text{mem}}$ points from the history of length $t = iL$. Its entries are defined by evaluating the Legendre basis functions at the chosen sample points $\{x_j\}_{j=0}^{L_{\text{mem}}-1}$:

$$(\mathbf{R}_i)_{jn} = g_n^{(iL)}(x_j) = \sqrt{2n+1} P_n \left( \frac{2x_j}{iL} - 1 \right). \tag{7}$$

At the beginning of processing block $i$, the memory Keys and Values are retrieved in parallel by looking up the appropriate reconstruction matrix $\mathbf{R}_{i-1}$ and performing a matrix multiplication:

$$\mathbf{K}_{\text{mem}} = \mathbf{R}_{i-1} \mathbf{C}_{i-1}^{(K)}, \qquad \mathbf{V}_{\text{mem}} = \mathbf{R}_{i-1} \mathbf{C}_{i-1}^{(V)}. \tag{8}$$

**Fixed-Length Polynomial Sampling.** A key innovation of our work is the strategic selection of the $L_{\text{mem}}$ sample points $\{x_j\}$ that define each reconstruction matrix $\mathbf{R}_i$. This process, which we term *Polynomial Sampling*, determines which parts of the history are retrieved. We explore two strategies:

- **Uniform Sampling (*Elastic Memory$_{uni}$*):** The sample points are uniformly spaced across the history $[0, iL)$, defined as $x_j = j \cdot \frac{iL}{L_{\text{mem}}}$ for $j = 0, \ldots, L_{\text{mem}} - 1$. This treats all parts of the history as equally important.

- **Exponential Sampling (*Elastic Memory$_{exp}$*):** The sample points are concentrated near the present and become exponentially sparser into the past. This is motivated by the linguistic intuition that recent context is most critical for immediate prediction, while distant history provides broader context.

**Integration via Trapezoidal Attention.** The retrieved memory tokens are seamlessly integrated into the Transformer's attention mechanism. The retrieved $\mathbf{K}_{mem}$ and $\mathbf{V}_{mem}$ are prepended to the current block's Keys $\mathbf{K}_{curr}$ and Values $\mathbf{V}_{curr}$, forming augmented sequences $\mathbf{K}_{aug} = [\mathbf{K}_{mem}, \mathbf{K}_{curr}]$ and $\mathbf{V}_{aug} = [\mathbf{V}_{mem}, \mathbf{V}_{curr}]$. The attention scores for the current block's queries $\mathbf{Q}_{curr}$ are then computed as:

$$\mathbf{S} = \frac{\mathbf{Q}_{curr}\mathbf{K}_{aug}^T}{\sqrt{D}}. \tag{9}$$

A trapezoidal attention mask $\mathbf{M}$ is applied to these scores. This mask allows all queries to attend to all $L_{mem}$ memory tokens, while enforcing causality within the current block. The final output from the attention computation is:

$$\mathbf{O}_{att} = \mathrm{softmax}(\mathbf{S} + \mathbf{M})\mathbf{V}_{aug}. \tag{10}$$

**Implicit Temporal Encoding.** Notably, the retrieved memory tokens do not receive external positional encodings. This design is intentional: the reconstruction process is *inherently time-aware*. As formalized in Equation 7, the reconstruction matrix $\mathbf{R}_i$ is constructed by evaluating Legendre basis functions at specific temporal coordinates $\{x_j\}$. Since the Legendre polynomial values $P_n(x_j)$ are unique, non-linear functions of the temporal coordinate $x_j$, the position information is mathematically intrinsic to each reconstructed vector. The retrieved key $\mathbf{K}_{mem}^{(j)}$ represents the reconstructed signal value $f(x_j) \approx \sum_n c_n P_n(x_j)$, where the temporal position is encoded through the basis coefficients rather than through additive positional embeddings.

# 3 EXPERIMENTS

To evaluate the performance of Elastic Memory, we conduct a comprehensive set of experiments on long-document language modeling. We structure our evaluation to assess several key dimensions: (1) core language modeling performance against state-of-the-art memory baselines across diverse domains; (2) the scalability of our model with respect to both memory capacity and model parameter size; (3) in-depth analyses of the memory mechanism itself, probing its flexibility and robustness; and (4) the computational efficiency of our method during training. The experimental results show that Elastic Memory achieves strong performance across all evaluation criteria, validating its effectiveness as a efficient, scalable, and theoretically-grounded memory architecture.

## 3.1 EXPERIMENTAL SETUP

**Datasets.** We evaluate our model on three challenging long-document datasets from diverse domains to test for generalization: **PG-19** (Rae et al., 2020), a collection of long-form books; **Proof-Pile** (Azerbayev et al., 2023), a dataset of mathematical papers; and **FineWeb-Edu**, a high-quality educational subset of the FineWeb dataset (Penedo et al., 2024). To construct a clean benchmark for long-context modeling, we preprocess these datasets by first filtering for documents exceeding 32,768 tokens. We then segment these long documents into non-overlapping chunks of exactly 32,768 tokens, discarding any trailing partial chunks. This process yields three datasets, each containing approximately 2 billion tokens, where every sample is an integer multiple of 32k tokens, providing a controlled, real-world environment for studying long-range dependencies. We set the block size to 2,048 tokens for all experiments.

**Baselines.** We compare Elastic Memory with representative state-of-the-art memory architectures, as summarized in Table 1: **Memorizing Transformer** (Wu et al., 2022) for *external memory*, **Infini-Transformer** (Munkhdalai et al., 2024) for *associative memory*, and **Melodi** (Chen et al., 2025) for *summary tokens*. We also include a Llama 3 architecture baseline, denoted as **Transformer++**, as a memory-free reference. Due to challenges in reproducing prior work, we re-implemented all baselines to ensure a fair and controlled comparison. All models are built upon a unified Llama 3

Table 1: Recurrent memory model comparison. Memory size and additional trainable parameters are counted per layer. $H$ represents the number of attention heads.

| Model | Memory Size | Add. Params | Update | Retrieval |
|---|---|---|---|---|
| **MemTrans.** | $N_{\text{queue}} \times (d_{\text{key}} + d_{\text{value}}) \times H$ | $H$ | **FIFO queue** | **kNN+ cross-attn** |
| **InfiniTrans.** | $d_{\text{key}} \times d_{\text{value}} \times H$ | $H$ | **Hebb rule+ delta rule** | **linear-attn** |
| **Melodi** | $(N_{\text{mem.}} + N_{\text{sum.}}) \times d_{\text{value}} \times H$ | $N_{\text{sum.}} \times d_{\text{value}} \times H$ | **self-attn+ linear-mix.** | **cross-attn** |
| **ElasticMem.** | $N_{\text{HiPPO}} \times (d_{\text{key}} + d_{\text{value}}) \times H$ | **0** | **function approx.** | **poly. sampling+ cross-attn** |

architecture, with modifications confined to the attention or memory mechanisms within the designated layers. This approach minimizes implementation differences and ensures that performance variations primarily reflect the efficacy of the memory systems themselves. Our implementations include slight modifications for fairness and stability: for Memorizing Transformer, we replace the original kNN retrieval with an end-to-end trainable dense attention mechanism, a modification also adopted by Chen et al. (2025); for Infini-Transformer, we add RMSNorm to stabilize training; for Melodi, we implement only its intra-layer memory propagation to align with the design of other baselines.

**Training.** To better isolate the effects of their memory mechanisms, all models are trained from scratch. We use a consistent training setup for all experiments, including a Llama 2 tokenizer, identical hyperparameters, and the same data sequence for each optimization step. All models are trained for **40 billion tokens**. If training instability such as loss spikes was observed, the experiment was restarted with the number of warmup steps doubled until training stabilized. To ensure a fair comparison, the baseline memory size (1x) for all models is aligned with that of Infini-Transformer, whose capacity is tied to its head dimension. We follow the memory injection strategies reported in the original papers: Memorizing Transformer and our Elastic Memory are injected into the 9th layer, while Infini-Transformer and Melodi are applied at every layer. We evaluate models using two metrics: validation loss, computed block-wise, and test Perplexity (PPL), computed using a sliding-window approach to better reflect performance on contiguous text.

Table 2: Language modeling results on PG-19, Proof-Pile, and FineWeb-Edu. Samples across all the datasets contain more than 32k tokens. Base model contains 100M parameters. Memory size and additional parameters are counted for the whole model.

| Model | Mem. Size | Add. Params | PG-19 | | Proof-Pile | | FineWeb-Edu | |
|---|---|---|---|---|---|---|---|---|
| | | | loss | PPL | loss | PPL | loss | PPL |
| **Transformer++** | 0 | 0 | 2.561 | 11.232 | 1.172 | 3.322 | 2.471 | 12.897 |
| **Mem. Transformer** | 0.8M | 96 | 2.546 | 11.050 | 1.166 | 3.305 | 2.473 | 12.924 |
| **Infini-Transformer** | 0.8M | 96 | 2.530 | 10.907 | 1.029 | 2.950 | 2.439 | 12.736 |
| **Melodi** | 0.8M | 1.5M | 2.512 | 10.684 | 1.031 | 2.940 | 2.431 | 12.509 |
| **Elastic Memory**$_{\text{uni}}$ | 0.8M | 0 | 2.509 | 10.692 | 1.026 | 2.958 | 2.424 | 12.419 |
| **Elastic Memory**$_{\text{exp}}$ | 0.8M | 0 | **2.508** | **10.651** | **1.018** | **2.915** | **2.419** | **12.318** |

## 3.2 COMPARING WITH BASELINES

**Setting and Results.** We first compare the main language modeling performance of Elastic Memory against the baselines. As shown in Table 2, our method consistently outperforms all baselines across the three diverse long-text datasets. Notably, *Elastic Memory*$_{exp}$ achieves the best results in all settings, demonstrating the effectiveness of its exponentially biased sampling. A key advantage of our approach is its design simplicity; Elastic Memory achieves these state-of-the-art results with

zero additional trainable parameters, which avoids introducing extra optimization burdens to the neural network, unlike methods such as Melodi which add trainable components to the memory.

Table 3: Memory scaling experiments on Proof-Pile. The base model contains 100M parameters. Numbers in parentheses indicate additional trainable parameters.

| Model | 1x Memory | 2x Memory | 4x Memory | 8x Memory | 16x Memory |
|---|---|---|---|---|---|
| **MemTrans.** | 3.31 (96) | 3.28 (96) | 3.12 (96) | 3.01 (96) | 2.94 (96) |
| **Melodi** | 2.94 (1.5M) | 2.93 (3.1M) | 2.88 (6.4M) | 2.84 (13.5M) | 2.84 (30.1M) |
| **Elastic.$_{uni}$** | 2.96 (0) | 2.91 (0) | 2.86 (0) | 2.81 (0) | 2.78 (0) |
| **Elastic.$_{exp}$** | **2.92** (0) | **2.86** (0) | **2.82** (0) | **2.78** (0) | **2.75** (0) |

### 3.3 SCALING UP MEMORY SIZE

**Setting and Results.** To evaluate the efficiency and scalability of our memory compression, we conduct experiments scaling the memory capacity from 1x to 16x on Proof-Pile. The results, presented in Table 3, highlight the remarkable compression efficiency of our approach. With just its baseline 1x memory size, *Elastic Memory$_{exp}$* already surpasses the performance of a 16x Memorizing Transformer. Furthermore, Elastic Memory consistently outperforms the powerful Melodi baseline, even when Melodi is allocated twice the memory capacity (e.g., 2x Elastic Memory outperforms 4x Melodi, and 4x outperforms 8x). This efficiency is particularly notable given the parameter overhead of Melodi, which grows substantially with memory size (adding over 30M parameters at 16x). In contrast, Elastic Memory maintains its performance edge with no additional trainable parameters, underscoring the effectiveness of its principled function approximation approach.

Table 4: Model scaling experiments on Proof-Pile, with model scales from 100M to 200M and 400M parameters. Infini-Transformer's memory size is tied with head dimensions, thus other memory models adjust their memory size to align with it.

| Model | 1x Params | | | 2x Params | | | 4x Params | | |
|---|---|---|---|---|---|---|---|---|---|
| | Mem. | loss | PPL | Mem. | loss | PPL | Mem. | loss | PPL |
| **Trans.++** | 0 | 1.172 | 3.322 | 0 | 1.143 | 3.226 | 0 | 1.139 | 3.206 |
| **MemTrans.** | 0.8M | 1.166 | 3.305 | 1.2M | 1.139 | 3.213 | 1.6M | 1.132 | 3.187 |
| **InfiniTrans** | 0.8M | 1.029 | 2.950 | 1.2M | 0.978 | 2.796 | 1.6M | 0.968 | 2.772 |
| **Melodi** | 0.8M | 1.031 | 2.940 | 1.2M | 0.999 | 2.840 | 1.6M | 0.979 | 2.771 |
| **Elastic.$_{uni}$** | 0.8M | 1.026 | 2.958 | 1.2M | 0.973 | 2.824 | 1.6M | 0.957 | 2.764 |
| **Elastic.$_{exp}$** | 0.8M | **1.018** | **2.915** | 1.2M | **0.967** | **2.775** | 1.6M | **0.955** | **2.737** |

### 3.4 SCALING UP MODEL SIZE

**Setting and Results.** We investigate whether the benefits of Elastic Memory persist as the base model size increases. We scale the underlying Transformer from 100M to 400M parameters, adjusting the memory size of all models accordingly to maintain a fair comparison. As shown in Table 4, Elastic Memory consistently maintains its performance advantage across all model scales. *Elastic Memory$_{exp}$* remains the top-performing model at 100M, 200M, and 400M parameter sizes, demonstrating that our memory mechanism is a robust component that effectively complements models of varying capacities.

### 3.5 INJECTING BIAS INTO MEMORY RETRIEVAL AT TEST-TIME

**Setting and Results.** A unique feature of Elastic Memory is the decoupling of its memory state representation from the retrieval mechanism. We test the flexibility of this design by training a model with one sampling strategy and evaluating it with another, without any retraining. The results are presented in Table 5 and Table 6. Most strikingly, the model trained with uniform sampling

Table 5: Bias injection experiments on PG-19, Proof-Pile, and FineWeb-Edu. We experiment with 4 polynomial sampling strategies during memory retrieval.

| Model | PG-19 | Proof-Pile | FineWeb-Edu |
|---|---|---|---|
| Elastic Memory$_{uni}$ | 10.692 | 2.958 | 12.419 |
| Elastic Memory$_{uni-exp}$ | 10.656 | 2.927 | 12.379 |
| Elastic Memory$_{exp}$ | 10.651 | 2.915 | 12.318 |
| Elastic Memory$_{exp-uni}$ | 10.696 | 2.965 | 12.519 |

Table 6: Bias injection experiments on Proof-Pile when scaling model size or memory size. We experiment with 4 polynomial sampling strategies during memory retrieval.

| Model | 1x | Scaling Memory | | | | Scaling Model | |
|---|---|---|---|---|---|---|---|
| | | 2x | 4x | 8x | 16x | 2x | 4x |
| Elastic Memory$_{uni}$ | 2.958 | 2.906 | 2.855 | 2.811 | 2.781 | 2.824 | 2.764 |
| Elastic Memory$_{uni-exp}$ | 2.927 | 2.881 | 2.836 | 2.797 | 2.769 | 2.799 | 2.749 |
| Elastic Memory$_{exp}$ | 2.915 | 2.860 | 2.816 | 2.777 | 2.749 | 2.775 | 2.737 |
| Elastic Memory$_{exp-uni}$ | 2.965 | 2.908 | 2.864 | 2.820 | 2.787 | 2.821 | 2.778 |

but evaluated with exponential sampling (*Elastic Memory$_{uni-exp}$*) consistently outperforms the model that was trained and evaluated with uniform sampling (*Elastic Memory$_{uni}$*). This demonstrates that our memory state learns a general, high-fidelity representation of the history, which can be queried more effectively by injecting a superior inductive bias (i.e., the exponential sampling strategy) at inference time. This result highlights the flexibility of our decoupled retrieval system.

Table 7: Local context corruption experiments on Proof-Pile under different corruption rates.

| Model | Origin. | Corrupted Rate | | | Hardly Corrupted Rate | | |
|---|---|---|---|---|---|---|---|
| | | 25% | 50% | 75% | 25% | 50% | 75% |
| Mem. Transformer | 3.305 | 3.312 | 3.316 | 3.316 | 3.956 | 5.118 | 34.392 |
| Elastic Memory$_{uni}$ | 2.958 | 2.991 | 3.024 | 3.319 | 3.553 | 4.554 | 30.529 |
| Elastic Memory$_{exp}$ | 2.915 | 2.930 | 2.948 | 3.316 | 3.475 | 4.367 | 30.575 |

## 3.6 PROBING MEMORY USAGE: CORRUPTED LOCAL CONTEXT

**Setting and Results.** To verify that our model genuinely relies on its long-term memory, we conduct an experiment where we intentionally corrupt the local context, thereby forcing the model to leverage historical information. We replace a percentage of the Key and Value tokens within the current block with random noise. We test two settings: "Corrupted," where noise is injected only in the memory-augmented layer, and "Hardly Corrupted," a more challenging setting where noise is injected in all layers. As shown in Table 7, while all models' performance degrades under corruption, Elastic Memory exhibits greater robustness, particularly in the more difficult "Hardly Corrupted" setting. This suggests that our memory mechanism provides a reliable source of information that the model can effectively fall back on when local context is compromised.

## 3.7 SPEED COMPARISON

**Setting and Results.** Finally, we evaluate the computational efficiency of Elastic Memory by measuring its training throughput, normalized against the Transformer++ baseline. The results in Table 8 show that our model's training speed is highly competitive. Its throughput is competitive with the Transformer++ and Memorizing Transformer baselines, and is significantly faster than both Infini-Transformer and Melodi. These results demonstrate that Elastic Memory's strong performance and scalability are achieved without compromising computational efficiency.

Table 8: Speed comparison when scaling model or memory size. Test the average data volume per second during training on Proof-Pile, normalized against Transformer++. Higher values are better.

| Model | 1x | Scaling Memory | | | | Scaling Model | |
| | | 2x | 4x | 8x | 16x | 2x | 4x |
|---|---|---|---|---|---|---|---|
| **Transformer++** | 1.000 | - | - | - | - | 0.880 | 0.579 |
| **Mem. Transformer** | 1.079 | 1.074 | 1.067 | 1.056 | 1.037 | 0.853 | 0.565 |
| **Infini-Transformer** | 0.637 | - | - | - | - | 0.515 | 0.357 |
| **Melodi** | 0.548 | 0.580 | 0.580 | 0.572 | 0.534 | 0.529 | 0.366 |
| **Elastic Memory$_{uni}$** | 1.055 | 1.050 | 1.032 | 0.978 | 0.795 | 0.847 | 0.555 |
| **Elastic Memory$_{exp}$** | 1.057 | 1.053 | 1.033 | 0.976 | 0.794 | 0.850 | 0.557 |

## 4 RELATED WORK

Our work builds upon a rich history of research in language model memory. We situate our contributions within three primary areas: external memory, associative memory, and summary tokens.

**External Memory.** A prominent line of work tackles fixed-context limits by augmenting Transformers with an external memory decoupled from the parameters, enabling a potentially vast repository of information. Transformer-XL (Dai et al., 2019) introduced segment-level recurrence, caching hidden states from previous segments to form a sliding window of short-term memory and extend context without recomputation. kNN-LM (Khandelwal et al., 2020) marked a shift to non-parametric memory: the model retrieves similar instances from a large external datastore to augment predictions, yielding long-term memory whose capacity scales with the datastore. Subsequent work refined this retrieval. Memorizing Transformer (Wu et al., 2022) integrated kNN retrieval via a differentiable attention layer, enabling end-to-end learning of when and how to use retrieved information. Recent efforts target retrieval efficiency and quality: Focused Transformer (Tworkowski et al., 2023) mitigates the "distraction issue"—where irrelevant memories dilute context—using contrastive learning, and LongMem (Wang et al., 2023) uses a decoupled side network to manage a memory bank while keeping the main LLM frozen. Despite strong performance, retrieval remains the bottleneck: as the store grows, low-latency, high-precision retrieval is hard, and maintaining and indexing large memories incurs notable computational overhead.

**Associative Memory.** Another vein reframes self-attention as an associative memory to tame quadratic compute and memory. Rooted in fast weight programming (Schmidhuber, 1992), the idea is to generate context-dependent "fast weights" on the fly, compressing history into the network state. DeltaNet (Schlag et al., 2021) shows linear attention as an efficient instantiation: memory accumulates key–value outer products to approximate full attention with linear complexity. Gated DeltaNet (Yang et al., 2025) adds LSTM-like gating to selectively forget outdated or irrelevant information. Infini-Transformer (Munkhdalai et al., 2024) incorporates associative memory into vanilla attention and uses the delta rule for more compressive updates, enabling infinitely long contexts with fixed memory—promising for streaming. A core limitation, however, is capacity: the outer-product state has fixed size (by model dimension and head count). As context grows, this constant-size memory compresses more aggressively, increasing interference and degrading fine-grained retrieval, forming a key bottleneck for truly scalable long-context modeling.

**Summary Tokens.** A complementary idea compresses history into a small set of summary tokens or vectors, avoiding full-history retention. The condensed representation is passed between segments. Block-Recurrent Transformers (Hutchins et al., 2022) add block-level recurrence with LSTM-style gating, propagating a compressed state. Recurrent Memory Transformer (Bulatov et al., 2022) appends learnable memory tokens per segment that serve as a dedicated "scratchpad," whose final states seed the next segment. AutoCompressor (Chevalier et al., 2023) learns fixed-size summary vectors as soft prompts for past segments, conditioning future predictions. Melodi (Chen et al., 2025) introduces hierarchical compression, producing short- and long-term summaries to reduce memory while preserving multi-scale context. The main challenge is summary quality: compression creates an information bottleneck, and effectiveness hinges on preserving salient information.

What counts as "salient" is task-dependent—a summary tuned for one objective (e.g., perplexity) may not transfer to others (e.g., QA)—hindering generalization.

## 5 CONCLUSION

We present Elastic Memory, a memory design that brings together theory and practical scale for long-context language models. Built on the HiPPO theory for online function approximation, it treats memory as optimal compression of history, with a clear update rule and a flexible polynomial-sampling retriever. It reaches state-of-the-art results on long-document benchmarks with zero extra parameters, while using memory well and scaling efficiently. Bias-injection tests show the benefit of the decoupled retriever, and context-corruption tests confirm real reliance on long-term history. These gains come with competitive training throughput, making the method practical. We trained from scratch to isolate the architecture; a next step is to test fine-tuning on large pretrained models.

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

# A  EXPERIMENTAL DETAILS

## A.1  TRAINING INFRASTRUCTURE AND HYPERPARAMETERS

All experiments were conducted on a single compute node equipped with 8 NVIDIA A800 GPUs. We provide the complete training configuration for reproducibility.

**Training Setup.** All models follow a unified Llama 3 architecture with modifications confined to the attention or memory mechanisms. Each optimization step processes approximately 0.5M tokens, with each sample containing 32,768 tokens. For memory-augmented layers, we input the full 32,768 tokens and process them in chunks of 2,048 tokens using the recurrent memory mechanism. All attention modules use the FlashAttention2 implementation for efficiency. The complete training hyperparameters are listed in Table 9.

Table 9: Training hyperparameters used in all experiments.

| Hyperparameter | Value |
|---|---|
| Optimizer | AdamW |
| Learning Rate | 2e-3 |
| Adam $\beta_1$ | 0.9 |
| Adam $\beta_2$ | 0.95 |
| Adam $\epsilon$ | 1e-7 |
| Weight Decay | 0.1 |
| Max Gradient Norm | 1.0 |
| LR Scheduler | Cosine |
| Warmup Ratio | 0.01 |
| Training Epochs | 20 |
| Batch Size per Device | 2 |
| Gradient Accumulation Steps | 1 |
| Context Length | 32,768 |
| Chunk Size | 2,048 |
| Precision | BF16 |
| Random Seed | 42 |

**Speed Measurement Methodology.** Table 8 reports training throughput normalized against Transformer++. We measure throughput using the `train_samples_per_second` metric reported by Weights & Biases. The raw (unnormalized) training throughput values are provided in Table 10.

Table 10: Raw training throughput (samples/second). Higher is better.

| Model | 1x | Scaling Memory | | | | Scaling Model | |
|---|---|---|---|---|---|---|---|
| | | 2x | 4x | 8x | 16x | 2x | 4x |
| **Transformer++** | 14.689 | - | - | - | - | 12.930 | 8.500 |
| **Mem. Transformer** | 15.844 | 15.774 | 15.677 | 15.514 | 15.237 | 12.531 | 8.297 |
| **Infini-Transformer** | 9.359 | - | - | - | - | 7.572 | 5.249 |
| **Melodi** | 8.052 | 8.523 | 8.518 | 8.403 | 7.851 | 7.777 | 5.370 |
| **Elastic Memory$_{uni}$** | 15.503 | 15.420 | 15.153 | 14.370 | 11.673 | 12.445 | 8.152 |
| **Elastic Memory$_{exp}$** | 15.530 | 15.471 | 15.179 | 14.331 | 11.663 | 12.491 | 8.177 |

**Inference Throughput.** To provide a comprehensive efficiency comparison, we also report inference throughput in Table 11, measured using the `eval/samples_per_second` metric from Weights & Biases. Elastic Memory maintains an efficiency advantage during inference, though the gap with baselines is narrower compared to training. We attribute this to the fact that Elastic Memory introduces no additional learnable parameters compared to the base model, which simplifies

the computational graph during training and yields a more pronounced efficiency benefit. During inference, where gradient computation is absent, this advantage is naturally reduced.

Table 11: Raw inference throughput (samples/second). Higher is better.

| Model | 1x | Scaling Memory | | | | Scaling Model | |
|---|---|---|---|---|---|---|---|
| | | 2x | 4x | 8x | 16x | 2x | 4x |
| **Transformer++** | 6.212 | - | - | - | - | 6.379 | 5.913 |
| **Mem. Transformer** | 6.608 | 6.544 | 6.463 | 6.523 | 6.455 | 6.287 | 5.873 |
| **Infini-Transformer** | 6.013 | - | - | - | - | 6.146 | 5.609 |
| **Melodi** | 5.552 | 5.558 | 5.646 | 5.738 | 5.653 | 5.698 | 5.078 |
| **Elastic Memory**$_{\text{uni}}$ | 6.406 | 6.512 | 6.457 | 6.428 | 6.106 | 6.322 | 5.798 |
| **Elastic Memory**$_{\text{exp}}$ | 6.498 | 6.480 | 6.444 | 6.367 | 6.126 | 6.338 | 5.848 |

**Memory Size Calculation.** The memory size metrics (1x, 2x, ..., 16x) reported in our experiments refer to the theoretical size of the recurrent memory module, as defined in Table 1. Specifically, the baseline memory size (1x) is aligned with that of Infini-Transformer, whose memory capacity equals $d_{\text{key}} \times d_{\text{value}} \times H$ per layer. For Elastic Memory, memory size scales with the HiPPO dimension $N$: at 1x, $N = 540$; at 2x, $N = 1080$; and so forth up to 16x with $N = 8640$.

## A.2 Context Length Coverage

A key distinction among memory models is the context length their memory can access:

- **Memorizing Transformer**: Uses a fixed-size KV queue (FIFO). The memory size directly determines context coverage. At 16x memory, it covers the full 32k context.
- **Infini-Transformer, Melodi, Elastic Memory**: These compressive memory models can access the *full 32k context* at all memory sizes. The memory size determines compression fidelity, not context coverage.

This distinction is crucial for interpreting Table 3: a 1x Elastic Memory accessing the full 32k context outperforms a 16x Memorizing Transformer that also covers 32k, demonstrating $16\times$ compression efficiency.

## B    MEMORY RECONSTRUCTION QUALITY

To validate that Elastic Memory faithfully preserves historical information, we analyze the reconstruction quality of the HiPPO-compressed memory states from two perspectives: synthetic signal reconstruction and language model Key/Value reconstruction.

### B.1    SYNTHETIC SIGNAL RECONSTRUCTION

We first validate HiPPO's compression capability on synthetic signals with known ground truth. We generate composite sine wave signals $f(t) = \sum_i A_i \sin(2\pi\omega_i t + \phi_i)$ with varying frequency compositions and compress them using HiPPO-LegS with different polynomial orders $N$ and discretization methods. The compressed state is then used to reconstruct the original signal, and we measure reconstruction MSE.

As shown in Table 12, HiPPO compression exhibits several properties:

- **Faithful low-frequency preservation**: Smooth sine wave signals achieve near-perfect reconstruction (MSE $\sim 10^{-5}$ to $10^{-4}$) even with small $N = 32$.
- **Robust degradation**: As signal complexity increases (more sine components), reconstruction quality degrades smoothly rather than catastrophically.

Table 12: HiPPO reconstruction quality on synthetic signals. Smooth signals (sine waves) achieve near-perfect reconstruction, while random noise cannot be compressed effectively.

| Signal Type | HiPPO Dim $N$ | Sample Length | Discretization | Reconstruct MSE |
|---|---|---|---|---|
| **1 sine wave** | 32 | 1024 | ZOH | 1.2e-5 |
| **3 sine waves** | 32 | 1024 | ZOH | 2.3e-4 |
| **5 sine waves** | 32 | 1024 | ZOH | 9.8e-4 |
| **5 sine waves** | 32 | 1024 | forward | 3.3e-3 |
| **5 sine waves** | 32 | 1024 | backward | 2.9e-3 |
| **5 sine waves** | 32 | 1024 | bilinear | 4.8e-4 |
| **random noise** | 32 | 1024 | ZOH | 0.97 |
| **random noise** | 128 | 1024 | ZOH | 0.90 |
| **random noise** | 512 | 1024 | ZOH | 0.72 |

- **High-frequency noise are incompressible**: Random noise, which can be treated as containing all frequencies, cannot be effectively compressed regardless of $N$.

These properties validate that HiPPO provides a principled compression mechanism suitable for preserving the smooth, low-frequency characteristics observed in language model hidden states.

### B.2 LANGUAGE MODEL KEY/VALUE RECONSTRUCTION

We further analyze reconstruction quality on actual Key/Value sequences from our trained models. We measure the Mean Squared Error (MSE) between the original Key/Value sequences and their reconstructions from the compressed memory state.

Table 13: Reconstruction quality vs. performance on Proof-Pile. As HiPPO dimension $N$ increases, reconstruction fidelity improves (lower MSE), corresponding to better language modeling performance (lower PPL).

| Memory Size | 1x | 2x | 4x | 8x | 16x |
|---|---|---|---|---|---|
| **HiPPO Dimension** $N$ | 540 | 1080 | 2160 | 4320 | 8640 |
| **Reconstruction MSE** | 0.45 | 0.41 | 0.35 | 0.31 | 0.27 |
| **PPL** | 2.92 | 2.86 | 2.82 | 2.78 | 2.75 |

The results in Table 13 confirm that increasing $N$ monotonically improves reconstruction fidelity without causing interference or overcompression. This validates that the polynomial basis provides a well-behaved compression mechanism where additional capacity translates directly to improved history representation.

## C  THEORETICAL GROUNDING: WHY HIPPO WORKS FOR LANGUAGE MODELS

A natural question arises: the HiPPO framework assumes continuous, smooth signals, yet language model hidden states are discrete sequences. We address this apparent mismatch.

Recent empirical analyses reveal that the deep representations of language models exhibit significant smoothness and low-frequency characteristics in the sequence dimension. Kai et al. (2025) analyzed the power spectrum of Llama-2-7B's Key/Value representations using DCT transforms and found that most energy concentrates in low-frequency components, with this concentration increasing in deeper layers. Similarly, He et al. (2023) observed analogous low-frequency dominance in Transformer hidden states.

These findings suggest that deep Transformer representations, despite originating from discrete tokens, exhibit the continuity properties that align with HiPPO's theoretical assumptions. Our design

choice to inject Elastic Memory at layer 9 (a deeper layer) is motivated by this observation. The strong empirical performance across diverse datasets validates this theoretical alignment.

## D  DISCUSSION: COMPRESSION VS. DIRECT SAMPLING

An intuitive alternative to our approach would be to directly sample tokens from the uncompressed history (e.g., uniformly selecting one token per block). We argue that HiPPO compression provides fundamental advantages over such direct sampling.

**Global Information Preservation.**   When projecting history onto the HiPPO basis, the resulting state $\mathbf{C}$ aggregates information from the *entire* trajectory. This process acts as a spectral low-pass filter, retaining the salient semantic structure while discarding high-frequency noise. In contrast, direct sampling suffers from information erasure: any token falling between sample points is completely lost (aliasing), creating blind spots in the model's memory.

**Empirical Validation.**   Our "Corrupted Context" experiments (Section 3.6) provide evidence for this advantage. Under local context corruption, Elastic Memory exhibits greater robustness than baselines, suggesting that the compressed representation provides a reliable information source that aggregates neighborhood information rather than point samples.

**Efficiency Comparison.**   The Memorizing Transformer represents an "upper bound" of raw storage (full FIFO queue). The fact that 1x Elastic Memory outperforms 16x Memorizing Transformer (Table 3) demonstrates that mathematically principled compression is more parameter-efficient than caching raw tokens.

## E  DETAILED MATHEMATICAL DERIVATIONS

This appendix provides detailed mathematical derivations for readers interested in a deeper understanding of the HiPPO framework and our Elastic Memory formulation. We expand upon the equations presented in Sections 2.2 and 2.3 of the main text.

### E.1  HIPPO-LEGS STATE SPACE DERIVATION

The HiPPO framework derives an Ordinary Differential Equation (ODE) that governs the evolution of the polynomial coefficients. We provide the key steps here.

**Step 1: Defining the Optimization Objective.**   At any time $t$, we seek a polynomial $g^{(t)}(x) = \sum_{n=0}^{N-1} c_n(t) g_n^{(t)}(x)$ that minimizes the approximation error:

$$\min_{c(t)} \int_0^t \left( f(x) - \sum_{n=0}^{N-1} c_n(t) g_n^{(t)}(x) \right)^2 \frac{1}{t} dx. \tag{11}$$

Since $\{g_n^{(t)}\}$ forms an orthonormal basis under the measure $\mu^{(t)}$, the optimal coefficients are given by:

$$c_n(t) = \langle f, g_n^{(t)} \rangle_{\mu^{(t)}} = \frac{1}{t} \int_0^t f(x) g_n^{(t)}(x) dx. \tag{12}$$

**Step 2: Deriving the Time Derivative.**   To obtain an incremental update rule, we differentiate $c_n(t)$ with respect to $t$. Using Leibniz's rule:

$$\frac{d}{dt} c_n(t) = \frac{d}{dt} \left[ \frac{1}{t} \int_0^t f(x) g_n^{(t)}(x) dx \right]$$

$$= -\frac{1}{t^2} \int_0^t f(x) g_n^{(t)}(x) dx + \frac{1}{t} f(t) g_n^{(t)}(t) + \frac{1}{t} \int_0^t f(x) \frac{\partial g_n^{(t)}(x)}{\partial t} dx. \tag{13}$$

The first term equals $-\frac{1}{t} c_n(t)$. The second term involves evaluating the basis at the boundary, where $g_n^{(t)}(t) = \sqrt{2n+1} P_n(1) = \sqrt{2n+1}$ since $P_n(1) = 1$ for all Legendre polynomials.

**Step 3: Computing the Basis Derivative.** The time derivative of the scaled Legendre basis is:

$$\frac{\partial g_n^{(t)}(x)}{\partial t} = \sqrt{2n+1} \cdot P_n' \left( \frac{2x}{t} - 1 \right) \cdot \left( -\frac{2x}{t^2} \right). \tag{14}$$

Substituting $z = \frac{2x}{t} - 1$, we get:

$$\frac{\partial g_n^{(t)}(x)}{\partial t} = -\frac{\sqrt{2n+1}}{t}(z+1)P_n'(z). \tag{15}$$

Using the Legendre polynomial derivative identity $(x+1)P_n'(x) = nP_n(x) + (2n-1)P_{n-1}(x) + (2n-5)P_{n-2}(x) + \cdots$, the integral of the basis derivative contribution becomes:

$$\frac{1}{t} \int_0^t f(x) \frac{\partial g_n^{(t)}(x)}{\partial t} dx = -\frac{1}{t} \sum_{k=0}^n \tilde{A}_{nk} c_k(t), \tag{16}$$

where $\tilde{A}_{nk}$ captures only the contribution from the basis derivative:

$$\tilde{A}_{nk} = \begin{cases} (2n+1)^{1/2}(2k+1)^{1/2} & \text{if } n > k \\ n & \text{if } n = k \\ 0 & \text{if } n < k \end{cases}. \tag{17}$$

Note that the diagonal element here is $n$, not $n+1$.

**Step 4: Combining All Contributions.** Recall that the first term from Step 2 contributes $-\frac{1}{t}c_n(t)$. When combined with the basis derivative contribution (which has diagonal element $n$), the total diagonal contribution becomes $n+1$. Thus, the final HiPPO-LegS matrices are:

$$\frac{d}{dt}c(t) = -\frac{1}{t}\mathbf{A}c(t) + \frac{1}{t}\mathbf{B}f(t), \tag{18}$$

where the HiPPO-LegS matrices are given by:

$$A_{nk} = \begin{cases} (2n+1)^{1/2}(2k+1)^{1/2} & \text{if } n > k \\ n+1 & \text{if } n = k \\ 0 & \text{if } n < k \end{cases}, \quad B_n = (2n+1)^{1/2}. \tag{19}$$

Note that $\mathbf{A}$ is lower triangular, which reflects the causal nature of the memory system. The diagonal element $n+1$ arises from combining the measure derivative contribution $(+1)$ with the basis derivative contribution $(n)$.

### E.2 DISCRETIZATION VIA ZERO-ORDER HOLD

To apply the continuous ODE to discrete sequences, we discretize using the Zero-Order Hold (ZOH) method.

**ZOH Discretization.** For a continuous-time linear system $\dot{c}(t) = A_c(t)c(t) + B_c(t)f(t)$ with time-varying coefficients, the ZOH discretization assumes $f(t)$ is constant over each discrete interval $[t_k, t_{k+1})$. The solution involves the state transition matrix $\Phi(t_2, t_1) = \exp\left(\int_{t_1}^{t_2} A_c(\tau)d\tau\right)$:

$$c(t_{k+1}) = \Phi(t_{k+1}, t_k)c(t_k) + \int_{t_k}^{t_{k+1}} \Phi(t_{k+1}, \tau)B_c(\tau)d\tau \cdot f_k. \tag{20}$$

**Application to HiPPO-LegS.** For HiPPO-LegS where $A_c(t) = -\frac{1}{t}\mathbf{A}$ and $B_c(t) = \frac{1}{t}\mathbf{B}$, we first compute the state transition matrix. With $t_k = k$ (unit time steps), we have:

$$\int_k^{k+1} A_c(\tau)d\tau = \int_k^{k+1} -\frac{1}{\tau}\mathbf{A}\,d\tau = -\mathbf{A}\ln\left(\frac{k+1}{k}\right). \tag{21}$$

Therefore, the discrete state matrix is:

$$\bar{\mathbf{A}}_k = \exp\left(-\mathbf{A}\ln\left(\frac{k+1}{k}\right)\right) = \left(\frac{k}{k+1}\right)^{\mathbf{A}}. \tag{22}$$

For the input matrix, we compute:

$$\bar{\mathbf{B}}_k f_k = \int_k^{k+1} \Phi(k+1, \tau) B_c(\tau) f_k \, d\tau = \int_k^{k+1} \left(\frac{\tau}{k+1}\right)^{\mathbf{A}} \cdot \frac{1}{\tau} \mathbf{B} f_k \, d\tau. \tag{23}$$

Evaluating this integral:

$$\begin{aligned}
\bar{\mathbf{B}}_k f_k &= \frac{1}{(k+1)^{\mathbf{A}}} \int_k^{k+1} \tau^{\mathbf{A}-\mathbf{I}} d\tau \cdot \mathbf{B} f_k \\
&= \frac{1}{(k+1)^{\mathbf{A}}} \cdot \mathbf{A}^{-1} \left[(k+1)^{\mathbf{A}} - k^{\mathbf{A}}\right] \mathbf{B} f_k \\
&= \mathbf{A}^{-1} \left[\mathbf{I} - \left(\frac{k}{k+1}\right)^{\mathbf{A}}\right] \mathbf{B} f_k.
\end{aligned} \tag{24}$$

Thus, the discrete input matrix is:

$$\bar{\mathbf{B}}_k = \mathbf{A}^{-1} \left(\mathbf{I} - \bar{\mathbf{A}}_k\right) \mathbf{B}. \tag{25}$$

Note that this follows from the standard ZOH formula for time-varying systems, where the sign is $(\mathbf{I} - \bar{\mathbf{A}}_k)$ rather than $(\bar{\mathbf{A}}_k - \mathbf{I})$ due to the negative sign in $A_c(t) = -\frac{1}{t}\mathbf{A}$.

These position-dependent matrices capture the time-varying dynamics of the HiPPO-LegS system.

### E.3 BLOCK-LEVEL PARALLELIZATION

We now derive the parallel block update formulation used in Elastic Memory.

**Unrolling the Recurrence.** Starting from the discrete recurrence $c_k = \bar{\mathbf{A}}_k c_{k-1} + \bar{\mathbf{B}}_k f_k$, we unroll over a block of $L$ steps:

$$\begin{aligned}
c_0 &= \bar{\mathbf{A}}_0 c_{-1} + \bar{\mathbf{B}}_0 f_0 \\
c_1 &= \bar{\mathbf{A}}_1 c_0 + \bar{\mathbf{B}}_1 f_1 = \bar{\mathbf{A}}_1 \bar{\mathbf{A}}_0 c_{-1} + \bar{\mathbf{A}}_1 \bar{\mathbf{B}}_0 f_0 + \bar{\mathbf{B}}_1 f_1 \\
&\vdots
\end{aligned}$$

$$c_{L-1} = \left(\prod_{k=L-1}^{0} \bar{\mathbf{A}}_k\right) c_{-1} + \sum_{j=0}^{L-1} \left(\prod_{k=L-1}^{j+1} \bar{\mathbf{A}}_k\right) \bar{\mathbf{B}}_j f_j. \tag{26}$$

**Matrix Formulation.** Define the state transition matrix $\mathbf{P} = \prod_{k=L-1}^{0} \bar{\mathbf{A}}_k$ and the HiPPO kernel $\bar{\mathbf{K}} \in \mathbb{R}^{N \times L}$ with columns:

$$\bar{\mathbf{K}}_{:,j} = \left(\prod_{k=L-1}^{j+1} \bar{\mathbf{A}}_k\right) \bar{\mathbf{B}}_j. \tag{27}$$

The block update then becomes a single matrix operation:

$$\mathbf{C}_{\text{new}} = \mathbf{P}\mathbf{C}_{\text{old}} + \bar{\mathbf{K}}\mathbf{F}, \tag{28}$$

where $\mathbf{F} = [f_0, f_1, \ldots, f_{L-1}]^T \in \mathbb{R}^{L \times d}$ is the input matrix.

### E.4 POLYNOMIAL RECONSTRUCTION

The memory retrieval leverages the reconstruction property of polynomial approximation.

**Reconstruction Formula.** Given the coefficient vector $c(t) \in \mathbb{R}^N$ at time $t$, the approximated signal value at any point $x \in [0, t]$ is:

$$\hat{f}(x) = g^{(t)}(x) = \sum_{n=0}^{N-1} c_n(t) g_n^{(t)}(x) = \sum_{n=0}^{N-1} c_n(t) \sqrt{2n+1} P_n \left( \frac{2x}{t} - 1 \right). \tag{29}$$

**Reconstruction Matrix.** To retrieve $L_{\text{mem}}$ sample points $\{x_0, x_1, \ldots, x_{L_{\text{mem}}-1}\}$ simultaneously, we construct the reconstruction matrix $\mathbf{R} \in \mathbb{R}^{L_{\text{mem}} \times N}$:

$$R_{jn} = \sqrt{2n+1} P_n \left( \frac{2x_j}{t} - 1 \right). \tag{30}$$

The reconstructed values are then obtained via matrix multiplication:

$$\hat{\mathbf{F}} = \mathbf{RC}, \tag{31}$$

where $\hat{\mathbf{F}} \in \mathbb{R}^{L_{\text{mem}} \times d}$ contains the reconstructed feature vectors at each sample point.

**Sampling Strategies.** The choice of sample points $\{x_j\}$ determines the reconstruction matrix and thus the information retrieved:

- **Uniform**: $x_j = j \cdot \frac{t}{L_{\text{mem}}}$ for $j = 0, 1, \ldots, L_{\text{mem}} - 1$.

- **Exponential**: $x_j = t \cdot (1 - \alpha^{L_{\text{mem}}-1-j})$ where $\alpha \in (0, 1)$ controls the decay rate.

Both strategies yield valid reconstructions; the choice depends on the desired attention pattern over history.

# F    THE USE OF LARGE LANGUAGE MODELS

We use LLMs to polish and assist in writing manuscripts. We did not use LLMs in any experiments (e.g., for evaluation). The core contributions, experimental design, and framework construction of the paper were all made by human researchers.

