# OpenReview forum: "Towards Compressive and Scalable Recurrent Memory"
_ICLR.cc/2026/Conference — Submitted to ICLR 2026_

### Official Review · Reviewer_RmKH · 2025-10-18

**Soundness:** 3
**Presentation:** 3
**Contribution:** 3
**Rating:** 6
**Confidence:** 3

**Summary:**

This paper proposes Elastic Memory, a theoretically grounded and scalable memory module for long-context Transformers. The method interprets historical tokens as samples from a continuous signal, compressing them into fixed-size memory states via optimal polynomial projection. The model retrieves contextual information using a precomputed reconstruction bank and a flexible polynomial sampling strategy, allowing retrieval biases to be injected at inference. Experiments on long-document benchmarks show improved perplexity and efficiency compared with baselines.

**Strengths:**

- The work elegantly unifies two previously competing paradigm: principled associative memory and scalable external memor, through the lens of online function approximation. Also, the paper provides concrete, closed-form derivations linking continuous HiPPO dynamics to discrete, parallelizable updates, offering strong conceptual clarity.

- The precomputation of HiPPO transition matrices transforms a theoretically sequential recurrence into an efficiently parallelizable block update, yielding strong training throughput without extra parameters.

- The experiments are comprehensive, demonstrating both performance and robustness.

- The ability to change polynomial sampling strategies at inference time without retraining is an interesting demonstration of decoupled memory representation and retrieval.

**Weaknesses:**

- The theoretical optimality of HiPPO relies on continuous, smooth signals under a uniform measure, which does not align with the discrete, non-stationary nature of token sequences in language modeling. Hence, the “optimal compression” claim is mathematically elegant but practically ungrounded.

- All evaluations focus on perplexity over long sequences. While lower PPL suggests improved context retention, it does not confirm enhanced semantic or logical long-term reasoning. No ablation or probing task demonstrates that Elastic Memory truly captures non-local dependencies beyond text continuity.

- Although the HiPPO framework provides a rigorous derivation, the system is trained end-to-end with gradient updates that can easily violate the theoretical assumptions. The authors present no evidence that the model actually maintains the intended HiPPO structure during optimization.

**Questions:**

- How sensitive is performance to the truncation order N of the polynomial basis? Does increasing N monotonically improve fidelity or cause overcompression/interference?

- What happens when the model processes sequences longer than the precomputed maximum context length? Is extrapolation possible without recomputation?

- Could the authors provide qualitative evidence showing that Elastic Memory indeed recalls semantically relevant distant context, rather than merely smoothing over text statistics?

---

> ### Author Response · Authors · 2025-11-24
> **Response to Reviewer RmKH (Part 1)**
>
> > Weakness 1: The theoretical optimality of HiPPO relies on continuous, smooth signals under a uniform measure, which does not align with the discrete, non-stationary nature of token sequences in language modeling. Hence, the “optimal compression” claim is mathematically elegant but practically ungrounded.
>
> You are right, the HiPPO framework is indeed derived under the assumption of continuous signals, and the smoother and more continuous the signal, the better the compression effect. However, the hidden states of language models on real data do exhibit a certain degree of smoothness and continuity in the sequence dimension, especially in the deeper layers of the model. For example, [1] analyzed the power spectrum of Llama-2-7b in the sequence dimension using DCT transform (see Figure 1), and it can be seen that most of the energy of Keys/Values is concentrated in the low-frequency part, and the concentration is higher in the deeper layers of the model; [2] performed a similar analysis on RoBERTa (see Figure 1), where the energy of hidden states is also concentrated in the low-frequency part and shows a trend of becoming more concentrated as the layer number increases. These two analytical works indicate that the deep representations of language models have low-frequency characteristics, which are consistent with the theoretical assumptions of the HiPPO framework to some extent. Our Elastic Memory inserts memory in the deeper layers of the model (layer 9) and has achieved good results in practice, confirming the above analysis.
>
> [1] FreqKV: Frequency Domain Key-Value Compression for Efficient Context Window Extension, arXiv 2025
>
> [2] Fourier Transformer: Fast Long Range Modeling by Removing Sequence Redundancy with FFT Operator, ACL 2023 Findings
>
> > Weakness 2: All evaluations focus on perplexity over long sequences. While lower PPL suggests improved context retention, it does not confirm enhanced semantic or logical long-term reasoning. No ablation or probing task demonstrates that Elastic Memory truly captures non-local dependencies beyond text continuity.
>
> We concur that evaluation on downstream tasks would strengthen our findings. Due to computational limitations, we leave the training of larger models required for meaningful downstream validation to future work. We followed the practices of Memorizing Transformer and Melodi, using PPL as the primary metric in this paper.

---

> ### Author Response · Authors · 2025-11-24
> **Response to Reviewer RmKH (Part 2)**
>
> > Weakness 3: Although the HiPPO framework provides a rigorous derivation, the system is trained end-to-end with gradient updates that can easily violate the theoretical assumptions. The authors present no evidence that the model actually maintains the intended HiPPO structure during optimization.
>
> We would like to clarify that the design of Elastic Memory actually preserves the HiPPO structure during training:
>
> *   As stated in Section 2.2, the HiPPO compression-related matrices (HiPPO kernel and state transition matrix) are pre-computed.
> *   Section 2.3 points out that the reconstruction bank used for the HiPPO recovery signal is also pre-computed.
> *   Finally, as shown in Algorithm 1, these matrices are dynamically loaded during the training process and are not updated by gradients.
>
> Therefore, we do not alter the HiPPO structure, preserving its original dynamics and mathematical properties.
>
> > Question 1: How sensitive is performance to the truncation order N of the polynomial basis? Does increasing N monotonically improve fidelity or cause overcompression/interference?
>
> In our model, the polynomial order N directly corresponds to the memory size (refer to Table 1 for the definition of memory size in our model). In Section 3.3, we systematically studied the impact of increasing the memory size, i.e., the value of N, on performance through experiments. As shown in Table 3, the model consistently and effectively utilizes the increased N to improve the fidelity of its historical representations, thereby achieving better performance. To more directly answer your question, we tested the MSE loss of Elastic Memory on the recovery of Keys/Values signals for different values of N, based on Table 3. The results are as follows. It can be seen that as N increases, the MSE loss continuously decreases, corresponding to better PPL results, and the model continuously gains from larger polynomial orders.
> | Memory size | 1x | 2x | 4x | 8x | 16x |
> | :--- | :--- | :--- | :--- | :--- | :--- |
> | **HiPPO dim N** | 540 | 1080 | 2160 | 4320 | 8640 |
> | **History reconstruct MSE** | 0.45 | 0.41 | 0.35 | 0.31 | 0.27 |
> | **PPL** | 2.92 | 2.86 | 2.82 | 2.78 | 2.75 |
>
> > Question 2: What happens when the model processes sequences longer than the precomputed maximum context length? Is extrapolation possible without recomputation?
>
> When the model processes a context that exceeds the pre-calculated length during inference, we can easily calculate A_k and B_k for each step beyond the predetermined length to correctly run HiPPO, which is similar to the RoPE recomputation when the model processes a context that exceeds the training phase length. We believe that the recurrence property of HiPPO determines its good performance during extrapolation.
>
> > Question 3: Could the authors provide qualitative evidence showing that Elastic Memory indeed recalls semantically relevant distant context, rather than merely smoothing over text statistics?
>
> Please refer to response to Weakness 2.

---

> ### Author Response · Authors · 2025-11-28
> **Response to Reviewer RmKH**
>
> Dear Reviewer RmKH,
>
> Thank you again for the constructive comments you gave us in your review. As the rebuttal phase will end on Dec 3, we would greatly appreciate it if you could also take some time to check if our rebuttal and revision has addressed your concerns, and please let us know if you would like us to provide any further clarification about the concerns you have.
>
> Best,
>
> Authors

---

### Official Review · Reviewer_tpqU · 2025-10-30

**Soundness:** 2
**Presentation:** 1
**Contribution:** 2
**Rating:** 2
**Confidence:** 3

**Summary:**

The authors introduce a technique called "elastic memory", which is based on the HiPPO framework.  The keys and values in a transformer are treated as samples from a continuous-time signal.  This signal is compressed into a finite memory C by fitting a polynomial approximation to the signal.  Crucially, updates to the memory are incremental and recurrent, so C can be updated with a new K,V pair on each time step.

The authors also derive a parallelizable block-level update operation, operating on blocks of length L.  Unfortunately, the block-level update operation, and the subsequent retrieval operations, depend on the absolute position of the block, so the authors pre-calculate and cache state-transition matrices and retrieval matrices up to some maximum context length.

To perform attention, the authors sample from the continuous signal at a pre-defined set of points; uniform sampling spreads these points uniformly across the input sequence, while exponential sampling concentrates the samples at recent positions.  The transformer then does dense attention over the samples.

**Strengths:**

The paper is well-written and clear, with well-defined algorithms and equations.

Unfortunately, I am not an expert in HiPPO, or state-space models in general, so it is hard for me to assess novelty and contribution.

**Weaknesses:**

The biggest weakness, as far as I am concerned, pertains to the writing style.  The method section presents a mathematical progression from HiPPO to elastic memory, but has almost no citations within it.  Thus, I cannot tell which parts of the math are novel derivations done by the authors, and which parts are drawn from prior literature.  I personally am not an expert in HiPPO or state space models, so I cannot determine contribution without citations.  The format of the equations differs enough from the HiPPO paper(s) that it is non-trivial to figure this out.

The second major weakness of the paper are the baselines.  There are two ways to measure memory against baselines.  The first is by the size of the memory (in number of parameters), the second is by context length that the memory covers.

The benefit provided by memory is largely measured in context length -- more context equals lower perplexity.  Memorizing transformer is extremely inefficient in its use of memory, so if you compare models based on the size of the memory, then memorizing transformer will obviously be worse when compared against anything that does compression -- a compressed memory model will *always* have more context for the same size of memory.  Thus, when used as a baseline, memorizing transformer should be used to measure the *quality* of compression -- a good compression mechanism should show little degradation over the uncompressed (memorizing transformer) baseline.

Given that the context length in this paper is 32k tokens, the memorizing transformer context length should be set to 32k.  Melodi similarly supports a maximum context length, although it uses a compressed representation.  Unfortunately, that is not what this paper does -- it uses a confusing 1x, 2x, 4x, 8x, 16x metric that I have no idea how to interpret.

Moreover, since this paper is based on HiPPO, and seems related to state-space models, it should be compared against some of the many models in the vast literature on linear transformers -- e.g. Mamba, RWKV-7, or DeltaNet.

Finally, memory models should ideally be tested on something other than perplexity -- ideally a downstream QA task, or at least a needle-in-a-haystack task.

**Questions:**

Which parts of the math are unique to this paper, and which parts are from prior literature?  Please list by equation (1) through (8).

The comparisons in Table 3 measure memory size as "1x, 2x, 16x," etc.  I have no idea what this metric means, and it makes no sense for Memorizing Transformer and Melodi -- both of which are models that measure the amount of "memory" in terms of context length.  What are the memory context lengths in Table 3?

What is trapezoidal attention?  Do you have a citation?

The retrieved keys and values do not have position encodings, because they are "inherently time aware" (line 228).  However, if C is indeed a "memory", then retrieving the memorized token at position x_j does not mean that the retrieved token has a timestamp incorporated into it.  Please explain.

Please double-check your references, e.g. I happened to notice that the authors on the block-recurrent transformers paper are incorrect.

The strategy of this paper seems to be:
(1) Store a compressed representation of the entire context.
(2) Sample a fixed (Lmem) number of samples from the memory, and then do dense attention.

Since attention is based solely on samples, couldn't you just sample directly from the original, uncompressed context, rather than compressing it first?  That would be much simpler and easier, e.g. uniform sampling would sample a few tokens from each block, and you could implement exponential sampling by gradually dropping tokens from more distant blocks.  I also would expect sampling without compression to perform poorly, which makes me wonder whether compression is doing something else here...

---

> ### Author Response · Authors · 2025-11-24
> **Response to Reviewer tpqU (Part 1)**
>
> > Weakness 1: The biggest weakness, as far as I am concerned, pertains to the writing style. The method section presents a mathematical progression from HiPPO to elastic memory, but has almost no citations within it. Thus, I cannot tell which parts of the math are novel derivations done by the authors, and which parts are drawn from prior literature. I personally am not an expert in HiPPO or state space models, so I cannot determine contribution without citations. The format of the equations differs enough from the HiPPO paper(s) that it is non-trivial to figure this out.
>
> Our Method section is divided into 3 parts: background, memory update, and memory retrieval, which are also the titles of the three subsections. Using "background" as the title clearly indicates that the content of this subsection refers to previous work. To minimize friction, we use the same mathematical notation as in the original HiPPO paper [1] to denote signals (f, g), coefficients (c), and HiPPO matrices (A, B), etc. For memory update and memory retrieval, these two subsections represent our core and original contributions. In the abstract and the first paragraph of Section 2, we explain that this paper designs a recurrent memory for language models based on the HiPPO framework, which includes two stages: update and retrieval. In summary, the mathematical framework of HiPPO is a contribution from previous work, and designing recurrent memory based on the HiPPO framework is our contribution. This is clearly stated in the abstract, the subsection titles, and the first paragraph of Section 2.
>
> > Weakness 2: The second major weakness of the paper are the baselines. There are two ways to measure memory against baselines. The first is by the size of the memory (in number of parameters), the second is by context length that the memory covers. The benefit provided by memory is largely measured in context length -- more context equals lower perplexity. Memorizing transformer is extremely inefficient in its use of memory, so if you compare models based on the size of the memory, then memorizing transformer will obviously be worse when compared against anything that does compression -- a compressed memory model will always have more context for the same size of memory. Thus, when used as a baseline, memorizing transformer should be used to measure the quality of compression -- a good compression mechanism should show little degradation over the uncompressed (memorizing transformer) baseline. Given that the context length in this paper is 32k tokens, the memorizing transformer context length should be set to 32k. Melodi similarly supports a maximum context length, although it uses a compressed representation. Unfortunately, that is not what this paper does -- it uses a confusing 1x, 2x, 4x, 8x, 16x metric that I have no idea how to interpret.
>
> * First, the calculation methods for the memory size of Elastic Memory and baseline are provided in Table 1, and the setting of memory 1x is described in lines 285-287. Correspondingly, 2x to 16x refer to the multiples by which the memory size is magnified. Lines 287-289 describe how to add memory modules to the base model. Combined with the above information, the metrics for 1x, 2x, ..., 16x memory can be understood.
> * Second, regarding the context length covered by memory, since the memory of Memorizing Transformer is a fixed-size KV queue, the only way to adjust its memory size is to adjust the queue length. For all other memory models, namely Infini-Transformer, Melodi, and Elastic Memory, the full 32k context can be accessed under all memory sizes in this paper.
> * Finally, regarding the performance comparison under different memory sizes: the memory queue of Memorizing Transformer with 16x already covers the full 32k context, but our model still surpasses it when using only one-sixteenth of the memory size (i.e., Elastic-exp-1x). This indicates that our model can maintain high quality while compressing the memory size by 16 times when accessing the full 32k context. On the other hand, Melodi can access the full 32k context under all memory sizes. Compared to Melodi, our model consistently achieves better performance. For example, 1x Elastic Memory can outperform 2x Melodi, 2x can beat 4x, and 4x can beat 16x. This means our memory compression efficiency has a significant advantage over Melodi.

---

> > ### Comment · Reviewer_tpqU · 2025-11-25
> >
> > I read lines 285-287 -- they state that: "baseline memory size (1x) for all models is aligned with that of Infini-Transformer,
> > whose capacity is tied to its head dimension" -- which explains exactly nothing, unless I want to chase down an external reference and puzzle out what you mean.
> >
> > While I appreciate the additional explanations you have given here, you seem to be more interested in defending the paper rather than updating it to address comments from my review.  Since most of my critiques still hold, I am not inclined to update my score.

---

> > > ### Author Response · Authors · 2025-11-26
> > > **Response to Reviewer tpqU**
> > >
> > > Dear Reviewer tpqU,
> > >
> > > Thank you for your thorough review and for pushing us to improve the paper's clarity. We have carefully considered all your feedback and made substantial revisions to address each concern.
> > >
> > > Revisions Made Based on Your Feedback
> > >
> > > 1. Contribution Clarity (Weakness 1)
> > >
> > >     We have revised the introduction of Section 2 to explicitly delineate prior work from our contributions:
> > >     - Equations 1-4 (Section 2.1 "Background: The HiPPO Framework"): From prior HiPPO literature [Gu et al., 2020]
> > >     - Equations 5-8 (Sections 2.2 "Memory Update" and 2.3 "Memory Retrieval"): Our novel contributions
> > >
> > >     Additionally, we have added Appendix E with detailed step-by-step mathematical derivations for readers unfamiliar with the HiPPO framework, including: (1) HiPPO-LegS state space derivation, (2) Zero-Order Hold discretization, (3) block-level parallelization, and (4) polynomial reconstruction formulas. Cross-references to these derivations have been added in Sections 2.1, 2.2, and 2.3. We apologize that the original structure, while might be clear to domain experts, was not sufficiently explicit for broader accessibility.
> > >
> > > 2. Memory Size Metrics (Weakness 2)
> > >
> > >     We have added Appendix A (Section A.1) with a detailed explanation of the memory size metrics:
> > >     - The 1x baseline is defined as Infini-Transformer's memory size: d_key × d_value × H per layer
> > >     - For Elastic Memory, this corresponds to HiPPO dimension N=540
> > >     - 2x through 16x represent proportional increases in N
> > >
> > >     We also clarify context length coverage in Section A.2: while Memorizing Transformer's 16x memory covers the full 32k context, our 1x model also accesses the full 32k context but with compressed representation—demonstrating 16× compression efficiency.
> > >
> > > 3. Position Encoding Explanation (Question 4)
> > >
> > >     We have expanded Section 2.3 with a new "Implicit Temporal Encoding" paragraph to explain that position information is mathematically intrinsic to reconstructed vectors: the Legendre polynomial evaluation P_n(x_j) at coordinate x_j makes the temporal position an inherent part of the reconstructed signal value, not a separate additive encoding. The detailed mathematical formulation is provided in Appendix E.4 (Polynomial Reconstruction).
> > >
> > > 4. Compression vs. Direct Sampling (Question 6)
> > >
> > >     We have added Appendix D addressing your insightful question. The key advantage of compression is global information preservation: HiPPO aggregates information from the entire trajectory (acting as a low-pass filter), while direct sampling loses all information between sample points. Our corruption experiments provide empirical evidence for this advantage.
> > >
> > > ---
> > >
> > > We genuinely appreciate your critical perspective, which has pushed us to make the paper more accessible to readers outside the SSM/HiPPO community. Your feedback has significantly improved the paper's clarity and completeness.
> > >
> > > We hope these revisions adequately address your concerns and demonstrate our commitment to producing high-quality, accessible research. We would be grateful if you could reconsider your evaluation in light of these substantial improvements.
> > >
> > > Thank you again for your time and valuable feedback. We remain open to addressing any remaining concerns you may have, and we are happy to make further revisions to improve the paper's quality.

---

> ### Author Response · Authors · 2025-11-24
> **Response to Reviewer tpqU (Part 2)**
>
> > Weakness 3: Moreover, since this paper is based on HiPPO, and seems related to state-space models, it should be compared against some of the many models in the vast literature on linear transformers -- e.g. Mamba, RWKV-7, or DeltaNet.
>
> Our work is indeed based on HiPPO, but it is orthogonal to a series of linear complexity sequence models such as state space models. This is because our model is designed to improve the long-context performance of Transformers by creating better recurrent memory, as demonstrated by the three baseline models we compared (all of which also added recurrent memory modules to Transformers, rather than completely abandoning them). The linear models you mentioned are orthogonal technical paths to Transformers, and neither of them involves the core of our work, which is the HiPPO matrix and the theory of online function approximation.
>
> > Weakness 4: Finally, memory models should ideally be tested on something other than perplexity -- ideally a downstream QA task, or at least a needle-in-a-haystack task.
>
> We appreciate your insightful suggestion and fully acknowledge the importance of evaluating Elastic Memory on downstream tasks. However, as such experiments require significantly larger pretrain models to obtain meaningful results, we have designated this investigation for future research. We followed the practices of Memorizing Transformer and Melodi, using PPL as the primary metric in this paper.
>
> > Question 1: Which parts of the math are unique to this paper, and which parts are from prior literature? Please list by equation (1) through (8).
>
> For the distinction between the contributions of this paper and prior literature, please refer to response to Weakness 1. Correspondingly, Formulas 1-4 in Section 2.1 are from prior work, while Formulas 5-8 in Sections 2.2 and 2.3 are our contributions.
>
> > Question 2: The comparisons in Table 3 measure memory size as "1x, 2x, 16x," etc. I have no idea what this metric means, and it makes no sense for Memorizing Transformer and Melodi -- both of which are models that measure the amount of "memory" in terms of context length. What are the memory context lengths in Table 3?
>
> Please refer to the response regarding Weakness 2.
>
> > Question 3: What is trapezoidal attention? Do you have a citation?
>
> We refer "trapezoidal" to the attention masking pattern. Lines 223-226 explain trapezoidal attention, where query tokens can access all memory tokens as well as their own KV tokens, forming an attention mechanism with a trapezoidal attention mask shape, as its name suggests.

---

> ### Author Response · Authors · 2025-11-24
> **Response to Reviewer tpqU (Part 3)**
>
> > Question 4: The retrieved keys and values do not have position encodings, because they are "inherently time aware" (line 228). However, if C is indeed a "memory", then retrieving the memorized token at position x_j does not mean that the retrieved token has a timestamp incorporated into it. Please explain.
>
> We clarify that "inherently time aware" refers to the mathematical definition of our retrieval mechanism as a function evaluation at specific temporal coordinates, rather than a retrieval of static stored embeddings. As formalized in Eq. 7 and Eq. 8, the reconstruction matrix $R$ is constructed by evaluating Legendre basis functions explicitly at the sample points $\{x_j\}$, where $(R)\_{jn} \propto P_n(x_j)$. Consequently, the retrieved key $K_{mem}$ is the reconstructed signal value $f(x_j) \approx \sum c_n P_n(x_j)$. Since the basis values $P_n(x_j)$ are unique, non-linear functions of the coordinate $x_j$, the temporal position is mathematically intrinsic to the reconstructed vector's value (via the basis coefficients), rendering explicit positional encodings redundant.
>
> > Question 5: Please double-check your references, e.g. I happened to notice that the authors on the block-recurrent transformers paper are incorrect.
>
> Thank you for pointing this out. We have checked and corrected the metadata of the cited literature.
>
> > Question 6: The strategy of this paper seems to be: (1) Store a compressed representation of the entire context. (2) Sample a fixed (Lmem) number of samples from the memory, and then do dense attention. Since attention is based solely on samples, couldn't you just sample directly from the original, uncompressed context, rather than compressing it first? That would be much simpler and easier, e.g. uniform sampling would sample a few tokens from each block, and you could implement exponential sampling by gradually dropping tokens from more distant blocks. I also would expect sampling without compression to perform poorly, which makes me wonder whether compression is doing something else here...
>
> * This is an insightful question that touches on the fundamental trade-off between time-domain sampling and spectral approximation. We believe the "something extra" provided by the compression mechanism is the preservation of global information density. When we project the history onto the HiPPO basis, the resulting state aggregates information from the *entire* continuous trajectory . This process effectively acts as a spectral low-pass filter, retaining the salient semantic structure of the whole history while discarding high-frequency noise.
> * In contrast, direct sampling from the raw context, even with an exponential schedule, inevitably suffers from information erasure: any token falling between sample points is lost entirely (aliasing), which creates blind spots for the model. By reconstructing the history from the compressed state, our retrieved "samples" are actually function evaluations that implicitly encode data from their surrounding temporal neighborhoods. This likely explains the superior robustness we observed in the "Corrupted Context" experiments (Section 3.6) .
> * Furthermore, considering that the Memorizing Transformer effectively represents the "upper bound" of raw storage (maintaining a full FIFO queue), the fact that a 1x Elastic Memory outperforms a 16x Memorizing Transformer (Table 3) suggests that a mathematically compressed representation is indeed more parameter-efficient for capturing long-term dependencies than simply caching raw tokens.

---

> ### Author Response · Authors · 2025-11-28
> **Response to Reviewer tpqU**
>
> Dear Reviewer tpqU,
>
> Thank you again for the constructive comments you gave us in your review. To address your concerns, we have revised and improved the manuscript content, enhancing its readability for readers unfamiliar with the field. As the rebuttal phase will end on Dec 3, we would greatly appreciate it if you could also take some time to check if our rebuttal and revision has addressed your concerns, and please let us know if you would like us to provide any further clarification about the concerns you have.
>
> Best,
>
> Authors

---

### Official Review · Reviewer_Uopc · 2025-11-01

**Soundness:** 3
**Presentation:** 3
**Contribution:** 3
**Rating:** 6
**Confidence:** 3

**Summary:**

This paper introduces Elastic Memory, a novel memory architecture for transformers designed to efficiently handle long-context sequences by leveraging the HiPPO framework for online function approximation. Elastic Memory compresses historical sequence information into a fixed-size memory state and retrieves it using a flexible polynomial sampling mechanism. The method is evaluated on several long-context language modeling benchmarks, consistently outperforming state-of-the-art baselines such as Memorizing Transformer, InfiniTransformer, and Melodi in terms of both memory efficiency and computational speed. The architecture is notable for its decoupled design, allowing for test-time injection of inductive biases, and achieves these results without introducing additional trainable parameters.

**Strengths:**

(1) The use of the HiPPO framework provides a solid mathematical basis for memory compression, moving beyond heuristic or ad hoc designs.
(2) Elastic Memory demonstrates state-of-the-art performance across multiple long-context benchmarks, outperforming strong baselines in both accuracy and efficiency.
(3) The architecture achieves its gains without adding extra trainable parameters, making it attractive for practical deployment.
(4) The ability to inject inductive biases at test time via different sampling strategies is a unique and potentially impactful feature.

**Weaknesses:**

(1) While the experiments are comprehensive within the long-context language modeling domain, the method is not evaluated on other important tasks such as vision or multimodal settings (e.g., vision-language models), which typically need long context window due to large number of image tokens from high-resolution images.

(2) The method is evaluated on models trained from scratch using metrics such as loss and ppl; its effectiveness when integrated into specific down-stream tasks remains to be demonstrated. For example, it would be great if the authors can run experiments for tasks such as long document summarization, which is a good use case for memory compression.

**Questions:**

see my comments above

---

> ### Author Response · Authors · 2025-11-24
> **Response to Reviewer Uopc**
>
> > Weakness 1: While the experiments are comprehensive within the long-context language modeling domain, the method is not evaluated on other important tasks such as vision or multimodal settings (e.g., vision-language models), which typically need long context window due to large number of image tokens from high-resolution images.
>
> Thank you for your suggestion! This paper focuses on the recurrent memory design of language models. We will explore the possibility of extending it to VLMs in the future, which is a very promising direction. Our contributions on the fundamental memory mechanism are orthogonal to modality. The principles proven here on sequence modeling (1D) form the theoretical foundation for future 2D (image) extensions.
>
> > Weakness 2: The method is evaluated on models trained from scratch using metrics such as loss and ppl; its effectiveness when integrated into specific down-stream tasks remains to be demonstrated. For example, it would be great if the authors can run experiments for tasks such as long document summarization, which is a good use case for memory compression.
>
> Thank you for your suggestions. We agree that validation on downstream tasks is important. Due to computational resource limitations, we will explore training larger models in the future to obtain meaningful test results on downstream tasks. We followed the practices of Memorizing Transformer and Melodi, using PPL as the primary metric in this paper.

---

> ### Author Response · Authors · 2025-11-28
> **Response to Reviewer Uopc**
>
> Dear Reviewer Uopc,
>
> Thank you again for the constructive comments you gave us in your review. As the rebuttal phase will end on Dec 3, we would greatly appreciate it if you could also take some time to check if our rebuttal and revision has addressed your concerns, and please let us know if you would like us to provide any further clarification about the concerns you have.
>
> Best,
>
> Authors

---

### Official Review · Reviewer_QUUx · 2025-11-01

**Soundness:** 2
**Presentation:** 3
**Contribution:** 2
**Rating:** 4
**Confidence:** 3

**Summary:**

Elastic Memory frames recurrent memory as an online function-approximation problem using the HiPPO (Legendre) projection and introduces a polynomial sampling reconstruction to produce a history summary that plugs into attention. The paper reports strong long-context results (32k+ sequences), claiming large memory efficiency and speed advantages over Memorizing Transformer and Melodi while keeping parameter counts comparable.

**Strengths:**

- Casting memory as HiPPO-based online compression gives a clear mathematical objective (optimal incremental polynomial projection) rather than an ad-hoc summarization heuristic; this grounds architecture choices in prior theory.

- The paper reports state-of-the-art performance on multiple 32k+ datasets, beating the Memorizing Transformer by large memory-efficiency margins and outperforming Melodi across memory sizes (including when Melodi has more parameters). These are high-impact claims if robust.

- The architecture decouples memory size from model dims and reports substantial runtime advantages when scaling model size (e.g., being faster than Melodi at 4× scale), which addresses an important production constraint.

**Weaknesses:**

- The HiPPO-projection order/size (N), the polynomial sampling schedule, and the weighting/measure choices can materially change performance. The paper gives strong aggregate wins, but I want systematic ablations (sensitivity to N, sampling density, training stability, and memory reconstruction quality metrics) to show results are robust and not brittle.

- The paper makes speed and memory-efficiency claims (e.g., 16× memory advantage; 50% faster at 4× scale) — but details on hardware, batch sizes, wall-clock measurement methodology, and training cost are thin.

- Show failure cases and limits: at what sequence lengths or signal types does the polynomial compression lose important information? How does it handle highly discontinuous or structured tokens (e.g., code, tables)?

**Questions:**

see weaknesses

---

> ### Author Response · Authors · 2025-11-24
> **Response to Reviewer QUUx (Part 1)**
>
> > Weakness 1: The HiPPO-projection order/size (N), the polynomial sampling schedule, and the weighting/measure choices can materially change performance. The paper gives strong aggregate wins, but I want systematic ablations (sensitivity to N, sampling density, training stability, and memory reconstruction quality metrics) to show results are robust and not brittle.
>
> *   Detailed observation! Regarding the HiPPO-related hyperparameters and settings you mentioned, we conducted systematic experiments and concluded that the model's use of HiPPO is robust and meets expectations:
>     *   For HiPPO projection size N: As shown in Table 1, we actually changed the memory size of Elastic Memory by increasing N in the experiments in Section 3.3. The good performance of Elastic Memory in Table 3 confirms that Elastic Memory can be trained stably across five different values of N.
>     *   For sampling density: In all experimental tables of this paper (Tables 2-8), we have simultaneously presented the performance of Elastic Memory under two sampling densities (uniform and exponential). The model performance under both sampling densities is excellent, outperforming most baselines, which indicates that Elastic Memory is robust to the choice of sampling density. Specifically, we also discussed the impact of using different sampling densities during training and testing on model performance in Section 3.5. We found that using exponential sampling density during testing can lead to performance gains, opening the door for injecting bias during testing.
>     *   For memory reconstruction metrics: We tested the ability of Elastic Memory to recover historical signal sequences for different values of N in Table 3, measured by the MSE loss between the reconstructed and true signals. The results are as follows: as N increases, the ability of Elastic Memory to recover history improves steadily, while PPL steadily decreases, which is consistent with the expected properties of HiPPO. Meanwhile, Elastic Memory outperforms baselines of the same memory size across all N values, meaning at all tested reconstruction qualities, indicating that the model is highly robust to reconstruction quality.
>         | Memory size | 1x | 2x | 4x | 8x | 16x |
>         | :--- | :--- | :--- | :--- | :--- | :--- |
>         | **HiPPO dim N** | 540 | 1080 | 2160 | 4320 | 8640 |
>         | **History reconstruct MSE** | 0.45 | 0.41 | 0.35 | 0.31 | 0.27 |
>         | **PPL** | 2.92 | 2.86 | 2.82 | 2.78 | 2.75 |
>
> > Weakness 2: The paper makes speed and memory-efficiency claims (e.g., 16× memory advantage; 50% faster at 4× scale) — but details on hardware, batch sizes, wall-clock measurement methodology, and training cost are thin.
>
> Thank you for pointing this out. We are now reporting the details of the speed tests, which will also be added to the appendix of the paper.
>
> * First, as indicated in lines 281-285, all models follow the same training hyperparameters and consistent code implementation. Each optimization step includes about 0.5M tokens, and each sample is 32768 tokens. We adopt a similar implementation strategy to Infini-Transformer, replacing the attention part of specific layers in the standard Llama 3 architecture. For the replaced layers, we input 32768 tokens at once and then process them in chunks of 2048 using a recurrent memory mechanism. We use the flash attention implementation for the attention modules of all models.
> * Table 8 reports the token throughput during model training, which is the number of samples processed by the model per second, normalized with respect to the results of Transformer++. We further show the unnormalized raw results below, where we directly use the model training throughput reported by Wandb, i.e., the value of the `train_samples_per_seconds` field, higher is better.
>     | Model | 1x | 2x Mem. | 4x Mem. | 8x Mem. | 16x Mem. | 2x Params. | 4x Params. |
>     | :--- | :--- | :--- | :--- | :--- | :--- | :--- | :--- |
>     | **Transformer++** | 14.689 | - | - | - | - | 12.93 | 8.5 |
>     | **Memorizing Transformer** | 15.844 | 15.774 | 15.677 | 15.514 | 15.237 | 12.531 | 8.297 |
>     | **Infini-Transformer** | 9.359 | - | - | - | - | 7.572 | 5.249 |
>     | **Melodi** | 8.052 | 8.523 | 8.518 | 8.403 | 7.851 | 7.777 | 5.37 |
>     | **Elastic Memory (uni)** | 15.503 | 15.42 | 15.153 | 14.37 | 11.673 | 12.445 | 8.152 |
>     | **Elastic Memory (exp)** | 15.53 | 15.471 | 15.179 | 14.331 | 11.663 | 12.491 | 8.177 |
> * For the calculation of memory size, our claimed 16x advantage is based on the theoretical size of the model's recurrent memory module, as indicated in Table 1, rather than the actual GPU memory occupancy.
> * For the hardware platform, we used a single 8-GPU A800 compute node for all experiments.

---

> ### Author Response · Authors · 2025-11-24
> **Response to Reviewer QUUx (Part 2)**
>
> > Weakness 3: Show failure cases and limits: at what sequence lengths or signal types does the polynomial compression lose important information? How does it handle highly discontinuous or structured tokens (e.g., code, tables)?
>
> * Based on your suggestions, we explored HiPPO's reconstruction capability for input signals of different natures, n - sine waves means the superposition of n sine waves with different frequencies. The results are as follows. It can be seen that HiPPO has stronger reconstruction ability for continuous and smooth input signals, and requires a larger N to fit more complex sequences. Different discretization strategies also have certain differences, among which ZOH performs better. These results are consistent with the error bounds derived in the HiPPO paper [1] (see Proposition 6, Section 3). Simply put, HiPPO loses important information when facing highly varying signals.
>     | Signal type | HiPPO dim N | Signal sample length | Discretization | History reconstruct MSE |
>     | :--- | :--- | :--- | :--- | :--- |
>     | **1 - sine waves** | 32 | 1024 | ZOH | 1.2e-5 |
>     | **3 - sine waves** | 32 | 1024 | ZOH | 2.3e-4 |
>     | **5 - sine waves** | 32 | 1024 | ZOH | 9.8e-4 |
>     | **5 - sine waves** | 32 | 1024 | forward | 3.3e-3 |
>     | **5 - sine waves** | 32 | 1024 | backward | 2.9e-3 |
>     | **5 - sine waves** | 32 | 1024 | bilinear | 4.8e-4 |
>     | **random noise** | 32 | 1024 | ZOH | 0.97 |
>     | **random noise** | 128 | 1024 | ZOH | 0.90 |
>     | **random noise** | 512 | 1024 | ZOH | 0.72 |
> * However, according to the analysis in [2,3], deep representations of language models have good low-frequency characteristics: the hidden states of language models on real language data do exhibit a certain degree of smoothness and continuity in the sequence dimension, especially in the deeper layers of the model.
>     * For example, [2] analyzed the power spectrum of Llama-2-7b in the sequence dimension using DCT transform (see Figure 1), and it can be seen that most of the energy of Keys/Values is concentrated in the low-frequency part, and the concentration is higher in the deeper layers of the model;
>     * [3] performed a similar analysis on RoBERTa (see Figure 1), where the energy of hidden states is also concentrated in the low-frequency part and shows a trend of becoming more concentrated as the layer number increases.
>     * These two analytical works indicate that the deep representations of language models have low-frequency characteristics, which are consistent with the theoretical assumptions of the HiPPO framework to some extent. Our Elastic Memory inserts memory in the deeper layers of the model (layer 9) and has achieved good results in practice, confirming the above analysis.
> * For code and tabular data, the Proof-Pile dataset tested in Tables 2-7 already includes code and tabular data. As can be seen from the data description on Hugging Face, it includes "ArXiv.math, math textbooks, ..., Lean mathlib and other Lean repositories, ...". Our model performs well on the Proof-Pile dataset, confirming its ability to handle structured data. Downstream code and tabular tasks differ greatly from natural language, thus we will train larger language models (e.g., 7B) in the future to demonstrate meaningful results on them.
>
>     [1] HiPPO: Recurrent Memory with Optimal Polynomial Projections, NeurIPS 2020
>
>     [2] FreqKV: Frequency Domain Key-Value Compression for Efficient Context Window Extension, arXiv 2025
>
>     [3] Fourier Transformer: Fast Long Range Modeling by Removing Sequence Redundancy with FFT Operator, ACL 2023 Findings

---

> ### Author Response · Authors · 2025-11-28
> **Response to Reviewer QUUx**
>
> To further alleviate your concerns about the computational efficiency comparison of baselines, we present the inference throughput on top of the already demonstrated training throughput as follows (we directly report the value of Wandb's `eval/samples_per_second` field). As can be seen, Elastic Memory still has an efficiency advantage, but the gap with the baseline has narrowed compared to training. We believe this is because Elastic Memory does not introduce any additional parameters compared to the base model, which simplifies the computational graph during model training, thus having a more significant efficiency advantage. This gap is narrowed during inference.
>
> | Model | 1x | 2x Mem. | 4x Mem. | 8x Mem. | 16x Mem. | 2x Params. | 4x Params. |
> | :--- | :--- | :--- | :--- | :--- | :--- | :--- | :--- |
> | **Transformer++** | 6.212 | - | - | - | - | 6.379 | 5.913 |
> | **Memorizing Transformer** | 6.608 | 6.544 | 6.463 | 6.523 | 6.455 | 6.287 | 5.873 |
> | **Infini-Transformer** | 6.013 | - | - | - | - | 6.146 | 5.609 |
> | **Melodi** | 5.552 | 5.558 | 5.646 | 5.738 | 5.653 | 5.698 | 5.078 |
> | **Elastic Memory (uni)** | 6.406 | 6.512 | 6.457 | 6.428 | 6.106 | 6.322 | 5.798 |
> | **Elastic Memory (exp)** | 6.498 | 6.48 | 6.444 | 6.367 | 6.126 | 6.338 | 5.848 |
>
> Thank you again for your time! If you still have concerns about computational efficiency or other aspects, please let us know, and we will try our best to address your concerns and improve the quality of the manuscript.

---

> ### Author Response · Authors · 2025-11-28
> **Response to Reviewer QUUx**
>
> Dear Reviewer QUUx,
>
> Thank you again for the constructive comments you gave us in your review. As the rebuttal phase will end on Dec 3, we would greatly appreciate it if you could also take some time to check if our rebuttal and revision has addressed your concerns, and please let us know if you would like us to provide any further clarification about the concerns you have.
>
> Best,
>
> Authors

---

### Author Response · Authors · 2025-11-26
**Global Response to All Reviewers**

We sincerely thank all reviewers for their constructive feedback, which has helped us substantially improve the paper. We have carefully addressed each concern and made the following revisions to the manuscript:

1. Clarified Contribution Structure (Reviewer tpqU): Revised the introduction of Section 2 to explicitly delineate prior HiPPO background (Equations 1-4) from our novel contributions (Equations 5-8).

2. Added Experimental Details (Reviewers QUUx and tpqU): Added Appendix A with training hyperparameters, training infrastructure, raw throughput values, memory size calculation methodology, and context length coverage comparison.

3. Added Memory Reconstruction Analysis (Reviewer QUUx): Added Appendix B with both synthetic signal reconstruction experiments (sine waves) and language model Key/Value reconstruction MSE analysis, demonstrating HiPPO's faithful compression properties.

4. Added Theoretical Grounding Discussion (Reviewer RmKH): Added Appendix C discussing why HiPPO's continuous-signal assumptions align with language model representations.

5. Clarified Position Encoding Mechanism (Reviewer tpqU): Expanded Section 2.3 with "Implicit Temporal Encoding" paragraph explaining how position is mathematically encoded through Legendre polynomial evaluations.

6. Added Compression vs. Direct Sampling Analysis (Reviewer tpqU): Added Appendix D explaining advantages of HiPPO compression over direct sampling.

7. Added Detailed Mathematical Derivations (All Reviewers): Added Appendix E with step-by-step derivations of the HiPPO-LegS ODE, discretization via Zero-Order Hold, block-level parallelization, and polynomial reconstruction formulas. Cross-references to this appendix have been added in the main text (Sections 2.1, 2.2, and 2.3) to help readers unfamiliar with the HiPPO framework follow the mathematical details.

We believe these revisions address all major concerns raised by the reviewers. We remain open to addressing any remaining concerns, and we are happy to make further revisions to improve the paper's quality.

---

### Meta-Review · Area_Chair_FR8F · 2026-01-03

**Summary:**

The reviewers acknowledged the strength of this paper being:
1. showing promising results compared to existing methods in memory without extra parameters.
2.  well written and easy to follow.

In the meanwhile, the reviewers raised a few concerns including:
1. missing detailed specs regarding hardware.
2. more experiments to justify the experimental gains.
3. lack of evaluation metrics other than loss/perplexity.

**Reviewer Concerns:**

What's been addressed:
1. study of hyperparameters and the relative gains.
2. speed tests and experiments specs.
3. comparison with other models/methods.

What are still outstanding:
1. extending beyond using PPL.
2. clarification of details as pointed out by reviewer tpqU.

**Reviewer Scores:**

Based on the discussions during rebuttal, I think the reviewers will likely keep their scores.

---

### Decision · Program_Chairs · 2026-01-26

Reject